# The genomic landscape of relapsed infant and childhood *KMT2A*-rearranged acute leukemia

Louise Ahlgren[1], Mattias Pilheden[1], Helena Sturesson[1], Guangchun Song[2], Michael P. Walsh[2], Minjun Yang [1], Maud Maillard[3], Huanbin Zhao [3], Zhongshan Cheng[4], Varsha Singh[1], Anders Castor[5], Cornelis Jan Pronk [5], Hanne Vibeke Marquart [6,7], Birgitte Lausen [8], Pauline Schneider [9], Gisela Barbany [10], Katja Pokrovskaja Tamm [11], Jonas Abrahamsson[12], Olli Lohi [13], Linda Fogelstrand [14,15], Pablo Menendez[16,17,18,19,20], Rob Pieters [9], Jinghui Zhang [21], Karin Lindkvist-Petersson [22], Jun J. Yang [3], Tanja A. Gruber [23], Ronald W. Stam [9], Jing Ma[2] & Anna K. Hagström-Andersson [1,24] ✉

To study the mechanisms of relapse in *KMT2A*-rearranged (*KMT2A*-r) acute lymphoblastic (ALL) and acute myeloid leukemia (AML), we performed whole-genome and exome sequencing of infants and children with relapsed ALL/AML ($n = 36$), and longitudinal deep-sequencing of 257 samples in 30 patients. Somatic alterations in drug-response genes, most commonly in *TP53* and *IKZF1* (64%), were highly enriched in early relapse ALL (79%, 9-36 months after diagnosis), but rare in very early relapse ALL (<9 months, 9%). A marked chemotherapy-exposure signature was detected for mutations in early relapse ALL but not in very early ALL or AML relapse, in line with different mechanisms of relapse. Longitudinal analyses could track residual leukemia cells, clonal drug responses, and the upcoming relapse. These results highlight that *KMT2A*-r ALL and AML evade therapy differently and provide insights into the mechanisms of relapse in this highly lethal form of pediatric acute leukemia.

Acute leukemia is the most common pediatric cancer with a 5-year overall survival rate above 90% for acute lymphoblastic leukemia (ALL)[1] and almost 80% for acute myeloid leukemia (AML)[2]. However, ALL in infants, i.e., children aged 0-12 months, with *KMT2A*-rearrangements (*KMT2A*-r), still have a dismal prognosis with a 21-45% 6-year event-free survival (EFS) rate compared to 74% EFS in non-*KMT2A*-r ALL[3]. This poor prognosis extends to children above 1 year of age with *KMT2A*-r ALL (69% EFS)[4]. The underlying pathology behind these lower cure rates is not well understood. However, new therapies including Blinatumomab[5] and chimeric antigen CAR T-cell (CAR-T)[6] therapy will hopefully improve future survival.

*KMT2A*-r infant ALL comprises 4% of ALL in children and whole genome sequencing (WGS) has shown few additional mutations, but

that activating kinase-PI3K/RAS-mutations occur in 50% of cases[7,8]. Over 90% of infants reach clinical remission (CR)[3], but are prone to rapid relapse, with 90% of relapses occurring within 2 years from diagnosis and 66% within one year[3,5], and our understanding of the mechanisms driving relapse remains limited. Mutations in *CREBBP*, *FPGS*, *IKZF1*, *MSH2/6*, *NR3C1/2*, *NT5C2*, *PRPS1/2*, *TP53*, and *WHSC1* are enriched at relapse across genetic subtypes in childhood ALL (17% in very early, <9 months after diagnosis; 65% early relapse, 9-36 months; 32% late relapse ALL, >36 months)[9–11]. In relapsed childhood AML, mutations in *FLT3*, *WT1*, and *UPTF* are enriched at relapse[12,13].

To gain insights into the mechanisms of relapse and clonal dynamics during treatment, we studied a cohort of 36 relapsed *KMT2A*-r infant and childhood ALL/AML cases with WGS and whole exome

sequencing (WES), providing the largest cohort of pediatric *KMT2A*-r acute leukemia reported to date. In addition, we performed targeted deep-sequencing of 257 samples from 30 cases, including 14 cases from the relapse cohort, during the disease course. Mutational signature analysis suggested that chemotherapy was the primary cause of mutations in early relapse ALL but not in very early relapse ALL or AML. 79% of early relapse ALL gained alterations in drug-response genes at relapse, most commonly in *IKZF1* and *TP53* (64%). By contrast, such alterations were rare in very early relapse ALL (9%), in line with inherent resistance, emphasizing the different mechanisms of relapse depending on relapse time. In AML, the same mutational processes were active at diagnosis and relapse, suggesting that the cells that gave rise to relapse were dormant during therapy and evaded therapy differently than the ALL cells. Combined, our data highlight the different mechanisms of relapse in *KMT2A*-r ALL and AML. Finally, longitudinal sequencing uncovered unique clonal responses, providing the foundation for future therapeutic interventions.

## Results

### Genomic landscape of relapsed *KMT2A*-r ALL

To gain insights into mechanisms of relapse, we performed WGS and WES on 36 cases of relapsed *KMT2A*-r ALL (*n* = 25, age range 1 day–1.3 years, average 164 days) or AML (*n* = 11, age range 121 days-17.7 years, average 4.7 years) including 26 infants (24 ALL, 2 AML) and 10 children (1 ALL, 9 AML) (average coverage: WGS 43X, WES 140X, Fig. 1a, b, Supplementary Fig. 1a, b, and Supplementary Data 1, 2). The mutational burden of single nucleotide variants (SNVs), insertion and deletions (indels), structural variants (SVs), and copy number alterations (CNAs), increased from diagnosis to relapse (Supplementary Fig. 1c, d, and Supplementary Data 3-6). *KMT2A*-r ALL and AML had a similar mutational burden at diagnosis, while at relapse, ALL displayed a higher frequency of mutations (*p* = 0.031) (Supplementary Fig. 1d). Three patients were hypermutated at relapse and therefore excluded from statistical comparisons (P11, P17, P137). There was no correlation between the number of mutations and age at diagnosis, or to the number of gained mutations at relapse and relapse time, specific *KMT2A*-r or age (Supplementary Fig. 1e–h).

In ALL, 20 mutated genes in five pathways were identified at relapse, that primarily affected cases that relapsed 9 months after diagnosis (early relapse, based on Li et al.[11]) with cases that relapsed before 9 months or had refractory disease (very early), having a paucity of such alterations: B-cell maturation (very early 27%, early 64%), cell cycle (very early 27%, early 50%), glucocorticoid receptor signaling (very early 0%, early 21%), purine metabolism (very early 0%, early 21%), and kinase/PI3K/RAS-signaling (very early 18%, early 50%). Importantly, 79% of early relapse ALL (11/14) harbored *IKZF1* (*n* = 8), *TP53* (*n* = 6), *NT5C2* (*n* = 2), *CREBBP* (*n* = 1), *PRPS2* (*n* = 1), *WHSC1* (*n* = 1) or *NR3C1* (*n* = 1) alterations at relapse, while such alterations were rare in very early relapse ALL (*IKZF1 n* = 1, 9%, 1/11) (Fig. 1c–f, Supplementary Fig. 1i, 2a–d, and Supplementary Data 7). *TP53* and *IKZF1* were the most frequently altered genes (64%, 9/14 early relapse) and co-occurred in 56% suggesting genetic cooperativity, with a trend towards a lower overall survival if either one was altered (*p* = 0.091) (Supplementary Figs. 2e, 3, 4 and Supplementary Data 8). *TP53* and *IKZF1* alterations were mainly detected in cases with the *KMT2A::AFF1*-fusion (80%). All *TP53*-alterations were gained at relapse, and most had multiple such alterations (1-3) with three cases having a complete loss of *TP53* (Supplementary Data 7). Similarly, only 2/9 *IKZF1*-alterations could be detected in the paired diagnostic sample with one identified by PCR only (P58), and one case lacking a paired diagnostic sample (Supplementary Fig. 5a). In addition, alterations in genes implicated in purine metabolism (*NT5C2*$^{R39Q/R367Q}$, *PRPS2*$^{P320L}$) and glucocorticoid signaling (*CREBBP*$^{G119fs}$, *WHSC1*$^{E1099K}$, focal deletion of *NR3C1*), co-occurred with *IKZF1* or *TP53* in 4/6 cases (Fig. 1d). The *PRPS2*$^{P320L}$ has not been described before and

when analyzing the Alpha Fold model, it was in close proximity of R302 (Supplementary Fig. 5b). The R302 residue affects the stability of hexamer formation when mutated and thereby also PRPS2 activity[14]. However, knockout of wild-type *PRPS2* alongside with overexpression of *PRPS2*$^{P320L}$ revealed that *PRPS2*$^{P320L}$ had minimal effects on viability in the presence of 6-mercaptopurine (6-MP) (Supplementary Fig. 5c–h). Finally, *CDKN2A/B* deletions were detected in 2/3 ALLs with refractory disease, and 1/14 early relapse ALL had copy number neutral loss of heterozygosity of chromosome 9p with no accompanying mutation in *CDKN2A/B* or *PAX5* (Fig. 1d and Supplementary Data 7). Targeted deep-sequencing did not detect these mutations at diagnosis and inspection of the WGS reads failed to detect the *NR3C1* deletion, suggesting acquisition during treatment or beyond our level of detection (Supplementary Data 9). To validate our findings, we analyzed data from three studies[7,8,11], including data from 80 diagnostic *KMT2A*-r infant ALL cases of which 18 had a paired relapse. This showed that 6/12 (50%) of infants with early relapse, and none of the 6 infants with very early relapse, had such alterations at relapse (Supplementary Data 10). Further, *TP53*-alterations were rare at diagnosis (2/80 cases), with one of the cases having a very early relapse, but data from the paired relapse was not available.

At diagnosis, 32% of patients (7/22) harbored signaling mutations and 36% at relapse (9/25), and mutations were enriched at early relapse (50% versus 18% of very early). Diagnostic *FLT3*-mutations were enriched at early relapse (23%) and may mark cases with high relapse risk. Only two cases maintained their signaling mutation from diagnosis to relapse and in the remaining cases, they were gained at relapse. Subclonal signaling mutations (variant allele frequency, VAF < 0.3) were more common at diagnosis (63% versus 25% at relapse) (Supplementary Fig. 5i, j and Supplementary Data 7). Combined, 79% of early relapse ALL had at least one alteration in *TP53*, *IKZF1*, *CREBBP*, *NT5C2*, *WHSC1* or *NR3C1* at relapse and the general paucity of such alterations in very early relapse/resistant ALL (9%) suggest that their mechanisms of relapse are different.

### Genomic landscape of relapsed *KMT2A*-r AML

Mutations in five pathways were enriched at AML relapse; cell cycle, transcription, WNT-signaling, epigenetic and signaling with only six genes recurrently altered including deletion of 12p affecting *CDKN1B* and *ETV6* (*n* = 3), mutations in *CCND3* (*n* = 2), *WT1* (*n* = 2), *SETD2* (*n* = 2), and *FLT3* (*n* = 2) (Fig. 2a–d and Supplementary Fig. 6a–e, 7a). Cell cycle alterations affected around 40–50% of early and late relapse AML, respectively and included heterozygous 12p deletions, and mutations in *CCND3*, *DZIP*, and *TP53*. The targets for the 12p deletions have been suggested to be *CDKN1B*, a negative cell cycle regulator, and/or the ETS-transcription factor *ETV6*[15]. *WT1*-mutations were gained at late relapse (2/6, 33%) and detected in cases with rare *KMT2A*-r (*KMT2A::AFDN, KMT2A::ELL*), whereas early relapse harbored mutations in other WNT-signaling pathway genes (40%, *APC, CTNNB1*). Around 40-50% of early and late relapse harbored epigenetic mutations which mainly included epigenetic writers (*MYST4/KAT6B, PRDM16, SMYD2*, and *SETD2*). *SETD2* was also inactivated by a translocation in a 12p-deleted case, which led to an in-frame *SETD2::DCP1B* product, where the C-terminal part of SETD2 containing the critical SRI-domain fused with 12p; consistent with a truncating mutation (Fig. 2e). Alterations in transcription factor genes were enriched at early relapse (60% vs 33%) and included *ETV6* (12p-del), and mutations in *NFE2, POU2F2, AHRR*, and *SREBF2*.

Signaling mutations were more common at diagnosis than at relapse in AML (73% versus 36%). These mutations were more often subclonal at relapse (67%, 4/6) than at diagnosis (33%, 3/9) and cases with subclonal signaling mutations at relapse had more than one such mutation at different VAFs (Supplementary Fig. 7b–d and Supplementary Data 7). Additionally, while diagnostic *KRAS/NRAS* mutations were maintained at early relapse they were lost at late relapse, which

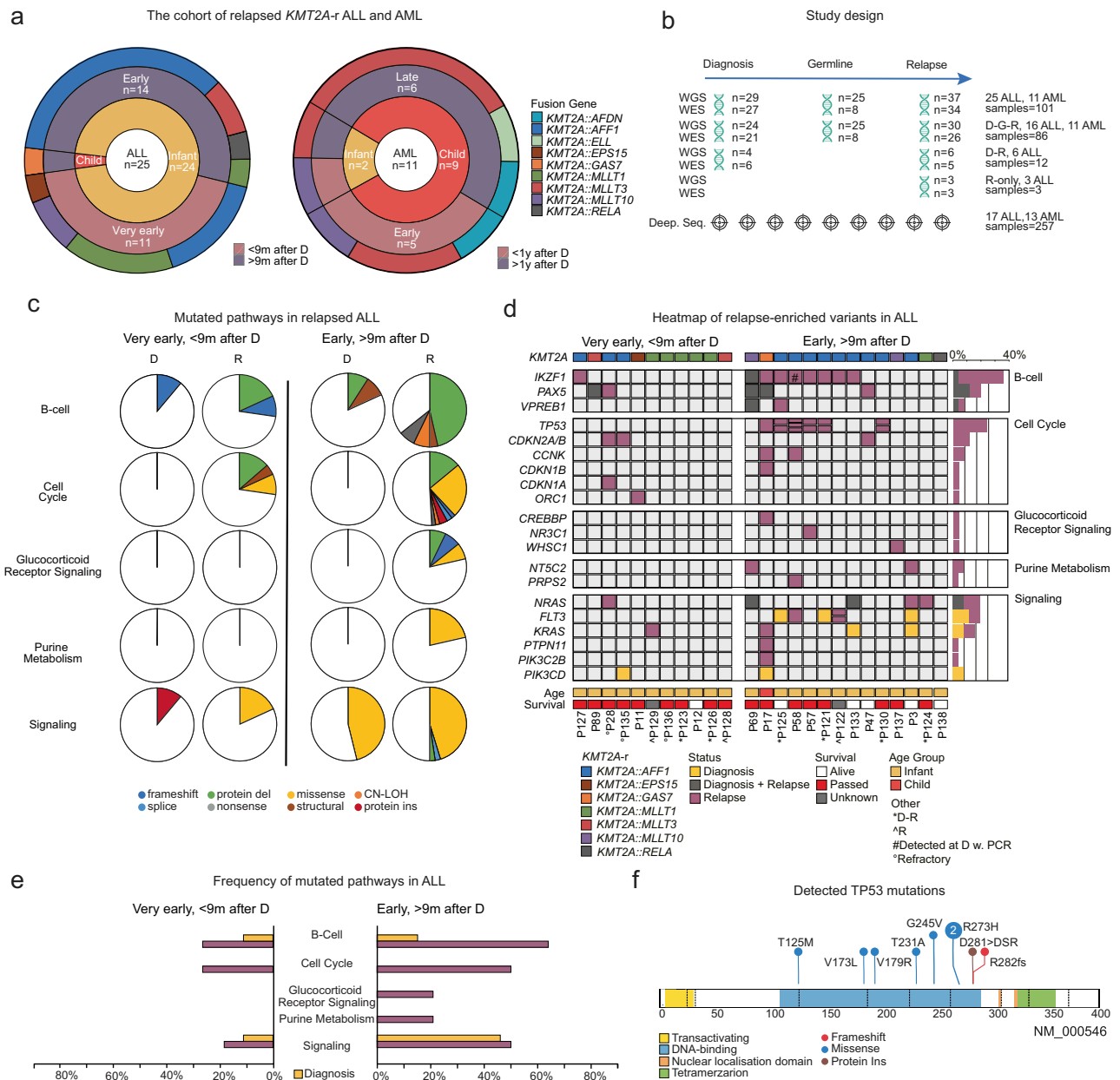

**Fig. 1 | The genetic landscape at relapse in *KMT2A*-r ALL. a** Illustration of the cohort divided into ALL and AML, age, relapse time, and specific *KMT2A*-r. ALL cases were divided into very early relapse (*n* = 11) if relapse occurred less than 9 months after diagnosis (D) or if the patient had resistant disease, and early relapse (*n* = 14) if relapse occurred after 9 months from diagnosis. The AML cohort was divided into early or late relapse if relapse occurred before (*n* = 5) or after 1 year after diagnosis (*n* = 6). **b** Schematic illustration of the study, including the number of AML and ALL patients and samples (D=Diagnosis, G=Germline, R=Relapse) analysed by each technology (whole genome, WGS; whole exome sequencing, WES). **c** Pathways mutated in very early and early relapse ALL. The colours indicate mutation type (CN-LOH, copy number loss of heterozygosity; protein del, protein deletion; protein ins, protein insertion), and the width of the slice, the number of

patients with a mutation in each pathway. **d** Heatmap of nonsynonymous mutations in genes within enriched pathways at relapse, with genes in rows and patients in columns. On the right, a bar plot indicating the fraction of patients with mutations in the specific gene, and on the left, the pathways, are shown. Yellow indicates diagnose-specific alterations, grey shared between diagnosis and relapse, and lilac relapse-specific. Note the *PTPN11^F28S.E8splice* (P17), is outside of the hot-spot sites, and although deleterious according to predictions, it is of unclear significance. D-R shows that no germline sample was available, R indicates that only a relapse sample was analysed. **e** The frequency of mutations in a certain pathway at diagnosis and relapse in very early and early relapse ALL. **f** Illustration of relapse-specific TP53-mutations, with the x-axis indicating the amino acid position, with our mutations at the top.

instead gained receptor tyrosine kinase mutations (*FLT3, CSF1R*). Within all enriched relapse pathways, early relapse AML maintained most mutations from diagnosis to relapse (60%, 12/20), whereas late relapse gained (38%, 8/21) and lost (29%, 6/21) more mutations (Fig. 2f). This suggests that early relapse AML, similar to very early relapse ALL, may exhibit an intrinsic drug resistance, whereas late relapse AML cases gain mutations that promote relapse.

## Mutational signatures at relapse fingerprints cellular history

We next analyzed all SNVs in non-repetitive regions showing that C > T transitions were the most common mutation at both diagnosis (45%) and relapse (44%), with no significant difference between ALL and AML (Supplementary Fig. 8a–c and Supplementary Data 11). Based on the trinucleotide context, 60 single-base substitution (SBS) signatures have been identified and attributed to both known and unknown etiologies[16].

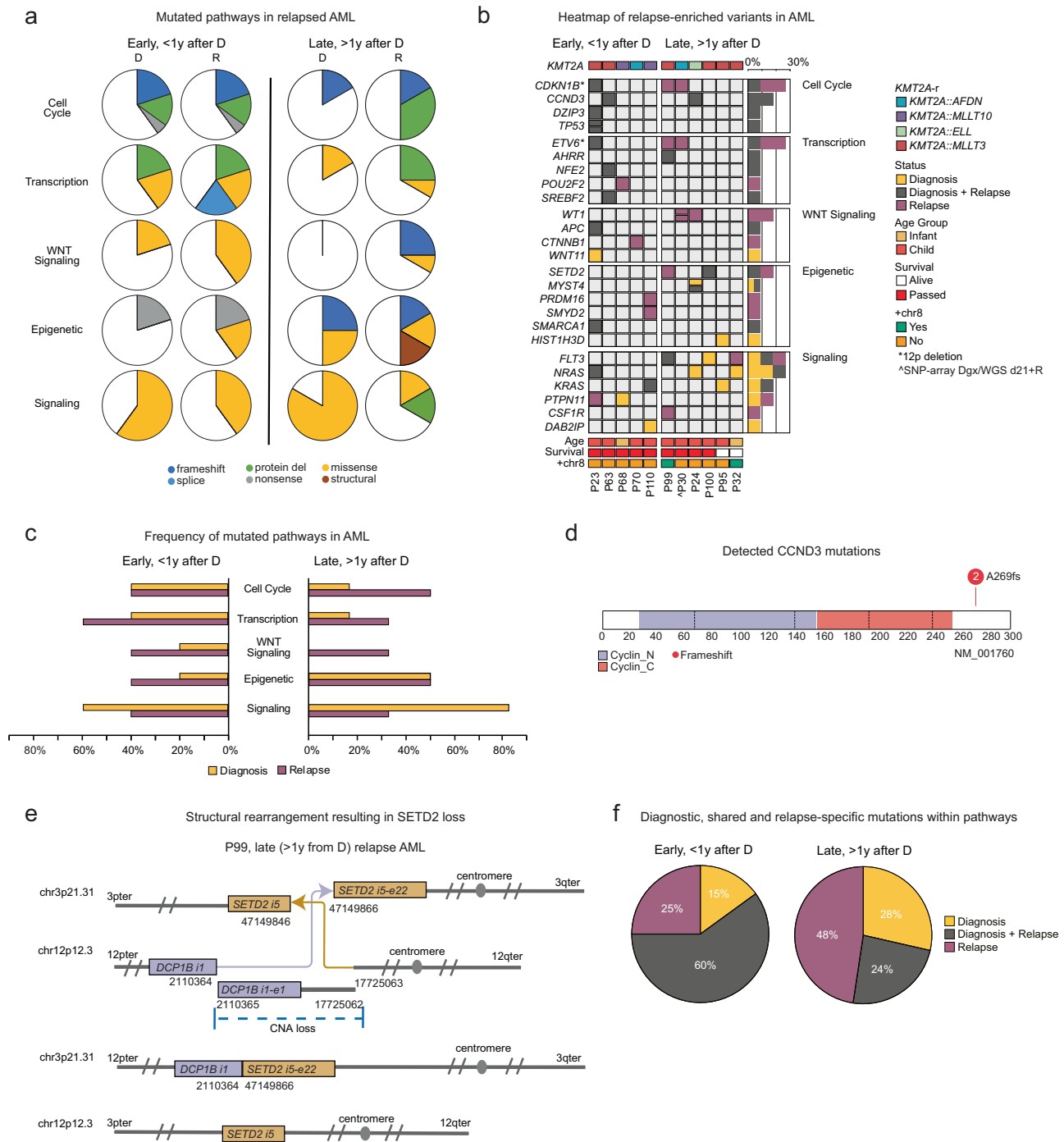

**Fig. 2 | Genomic landscape of relapsed *KMT2A*-r AML. a** Enriched pathways at relapse in early (*n* = 5, before 1 year after diagnosis) and late relapse (*n* = 6, after 1 year from diagnosis) *KMT2A*-r AML. The colors indicate mutation type, and the width of the slice, the number of patients with a mutation in each pathway at diagnosis (D) and at relapse (R). **b** Heatmap of nonsynonymous mutations in pathways enriched at relapse, with genes in rows and patients in columns. On the right, a bar plot indicating the fraction of patients with mutations in the specific genes, and on the left, the pathways are shown. Yellow indicates that the mutation is diagnose-specific, grey shared at diagnosis and relapse and lilac relapse-specific (gained). Note for P30, WGS was performed on a sample 21 days (estimated around 20–30% leukemia cells) from diagnosis in addition to the relapse samples. SNP-array data is included for the diagnostic sample. **c** The frequency of mutations in a certain pathway at diagnosis and relapse in early and late relapse AML. **d** Protein paint illustration of the CCND3 mutations, with the x-axis indicating the amino acid position, with our mutations shown at the top, fs frame shft. **e** Schematic picture of the structural rearrangements leading to SETD2 loss, chr chromosome, pter p terminal, qter q terminal. **f** Pie charts showing the frequency of gained, lost, and maintained mutations within the enriched pathways in early and late relapse AML.

To understand which mutational processes that were active, we examined the mutational signatures at relapse. In ALL, 9/14 cases had chemotherapy exposure as their primary signature and 36%-86% of their mutations were attributable to chemotherapy exposure (early relapse 89% vs very early 20%), particularly SBS87 which has previously been correlated to thiopurine treatment (Fig. 3a)[17]. Five cases did not have chemotherapy as their primary signature; in P28, P135, and P136 which had refractory disease, the primary signature was of unknown origin, with chemotherapy being the primary known signature for P28, and mismatch repair (MMR) for P135 and P136. P17 had MMR as the

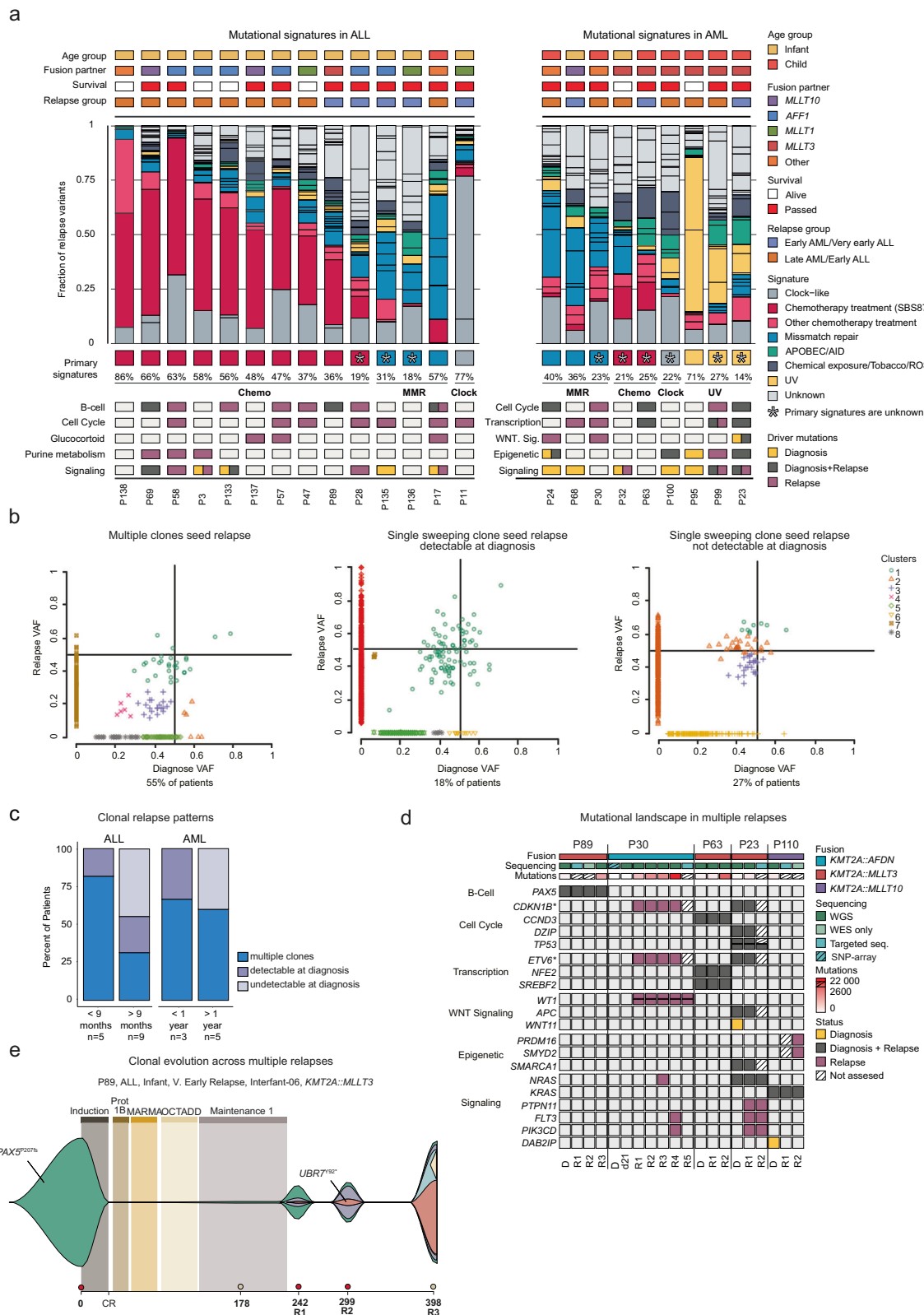

primary signature but exhibited a smaller subset of chemotherapy-associated mutations at an elevated VAF (Supplementary Fig. 8d). Finally, P11 was hypermutated at relapse, with 77% of mutations having a clock-like signature. Cases with mutations affecting purine metabolism (P69, P58 and P3), had the most prominent chemotherapy-induced signatures and very early relapse, the least prominent.

As opposed to ALL, the most common mutational signature at AML relapse was of unknown origin (6/9) with chemotherapy-associated signatures virtually absent (Fig. 3a). To investigate this further, we performed an analysis of relapse-specific mutations only and compared them to the diagnostic profiles, showing that in contrast to ALL, the cause of mutations at AML relapse did not differ from that

**Fig. 3 | Mutational signatures and clonal evolution patterns. a** Mutational signatures at relapse in ALL (left) and AML (right). Each patient is annotated based on age, *KMT2A*-r, treatment, survival, and relapse group. Each signature is classified into Chemotherapy treatment (mutations due to thiopurine exposure (SBS87) is highlighted), Mismatch repair (MMR), APOBEC/AID, UV, Clock-like, Miscellaneous (Chemical exposure/Tobacco/ROS/Bacterial) or Unknown. The most prominent known signature is denoted under the bar chart, with an asterisk if the most common signature was 'unknown'. Mutations in significantly mutated pathways are colored based on whether the mutation was found at diagnosis (yellow), relapse (lilac), or both (gray). Note, for P89, the third relapse was analyzed. **b** 2D plots showing the assignment of mutations to clusters at diagnosis and relapse, with each showing a representative patient with multiple clones seeding relapse, or with a clonal sweep that either was or was not detected at diagnosis, and at the bottom, the fraction of patients is stated. VAF=variant allele frequency. **c** The fraction of patients with multiple clones seeding relapse, or with a clonal sweep that was or was not detected at diagnosis, separated by leukemia type and when the patient relapsed. **d** Heatmap showing mutated genes in the indicated pathways in patients with multiple relapses. Annotated on top: *KMT2A*-r, sequencing type, and number of mutations. The number of mutations is only counted for samples that underwent WGS; Note for P30, WGS was performed on a sample 21 days (estimated around 15% leukemia cells) from diagnosis in addition to the relapse samples. SNP-array data is included for the diagnostic sample. **e** Clonal evolution depicted for P89 with multiple relapses, illustrated by a fish plot. At the top, the different treatment blocks are indicated, and at the bottom, the sampling day from diagnosis by tick marks. A circle above the tick mark shows if the sample was from BM (grey) or PB (red), CR complete remission, R1 first relapse, R2 second relapse, R3 third relapse. Samples sequenced by targeted sequencing, WGS, and WES are shown in bold. Non-silent mutations and their clonal relationship are indicated.

at diagnosis and thus mirrored the natural acquisition of mutations (AML Spearman $r^2 = 0.75$ vs ALL $r^2 = 0.36$) (Supplementary Fig. 8e). Moreover, the mutational processes in AML remained stable across multiple relapses (Supplementary Fig. 9a). Combined, early relapse ALL was characterized by mutations that associated with chemotherapy exposure, indicating that the surviving cells accumulate mutations because of cytotoxic cell damage. Relapsed AML lacked chemotherapy-exposure signatures suggesting that the relapse originated from an evolutionary early cell that remained unaffected by chemotherapy.

## Clonal evolution patterns from diagnosis to relapse

We next investigated the pattern of clonal evolution from diagnosis to relapse, showing that all relapses lost diagnosis-specific variants indicative of branching evolution (Supplementary Figs. 9b, 10a). By comparing the diagnostic and relapse VAFs, it was shown that in 55% of cases (12/22), relapse was seeded by multiple diagnostic clones, and in the remaining 45%, by a single sweeping clone that was detected at diagnosis in 40% of those cases (Fig. 3b, c, and Supplementary Data 3). Most patients had additional subclones at relapse, in line with a continuous evolution creating new clones. The lack of detection of the relapse clone at diagnosis does not exclude that it was present but below our resolution. In ALL, the clonal evolution pattern correlated to time to relapse with multiple diagnostic clones seeding relapse seen in 80% of very early and 33% of early relapse ALL, whereas in AML the frequency was the same (early, 66% and late, 60%) (Fig. 3c).

One ALL and four AML had multiple BM relapses, allowing us to study how the genetic landscape evolved across consecutive relapses (Fig. 3d, Supplementary Fig. 11a–d and Supplementary Data 3). The mutational burden increased after each relapse and mutations in driver genes were not lost when once gained at relapse suggesting a fitness advantage. This stepwise replacement of a fitter clone is illustrated in P89, an infant ALL with three relapses (Fig. 3e and Supplementary Fig. 11e). At the first relapse, the diagnostic *PAX5*[P207fs]-clone was dominant and a new subclone emerged in 22% of cells, which acquired a *UBR7*[Y92*] at the second relapse, gained new mutations and expanded in a sweep at the third relapse, where a new subclone emerged. Moreover, 64 days before the first relapse, the patient was in CR by flow cytometry (FC), but NGS detected the relapse clone in 2% of cells.

## Residual leukemia cells and unique clonal responses

To uncover clonal responses to treatment and study measurable disease at the clinical measurable residual disease (MRD) timepoints, we studied 19 relapse (10 ALL, 9 AML) and 11 remission (7 ALL, 4 AML) cases using patient-specific mutations, including the *KMT2A*-r, by deep-sequencing (average coverage 3250/site, average 17 mutations/patient, 257 samples) (Fig. 1b, Supplementary Figs. 12–16b, and Supplementary Data 1, 12–14). CR was defined as <5% blasts in patients treated on protocols from year 2000 or older and by FC < 0.1% for newer protocols. The VAFs between the WGS/WES and targeted gene

resequencing correlated ($r^2 = 0.66$) and a dilution series showed a sensitivity around VAF 0.002 or 0.4% of leukemia cells (Supplementary Fig. 16c–e).

At day 15, the first MRD-measurement during induction therapy, 13/15 ALL cases had measurable disease, with a similar average VAF in relapse and remission cases, whereas at the end of induction (EOI, day 29), relapse cases had a trend towards higher values ($p = 0.069$) (Fig. 4a, b and Supplementary Fig. 16f–j, 17a). This was mainly driven by very early relapse cases and if no mutations were detected, patients remained in remission. Cases with *KMT2A::AFF1* and rare *KMT2A*-r still had measurable disease at the EOI while those with *KMT2A::MLLT3* were negative (Supplementary Fig. 17b). A subset of cases had measurable disease at day 15 or 29 despite being in CR (Supplementary Fig. 17c, d and Supplementary Data 14).

Around half of AML cases had measurable disease after the first induction (EO1I) and the average VAF was similar in remission and relapse patients, with two having measurable disease despite being in CR (Fig. 4c and Supplementary Fig. 17e-h). After the second induction (EO2I, days 36-76), 7/12 cases had measurable disease with 3 cases being in CR by clinical MRD, and 6/7 cases relapsed (Fig. 4d and Supplementary Fig 17i). Further, 3/4 early relapse AML had higher values after EO2I as compared to EO1I, and 4/6 *KMT2A::MLLT3*-r cases had decreased values, concordant with their favorable outcome (Supplementary Fig. 17j, k).

Low-frequency *KMT2A*-fusion positive cells were found at CR outside of the MRD time points in 11/30 patients (3 remission, 27%; 8 relapses, 42%) (Supplementary Data 12). This included four infant ALLs with measurable disease across all time points until relapse (6-13 time points), one had resistant disease (P28), but the others entered CR (P12, P58, P89) (Figs. 3e, 4e and Supplementary Fig. 12a, c). Excluding these cases, in 4/16 cases with a BM relapse, we detected the relapse clone after induction therapy a long as 25–92 days before relapse. Some diagnostic mutations/SVs were always present and sufficient to detect the relapse. Combined, molecular monitoring with patient-specific mutations readily detected the leukemia cells.

The longitudinal data allowed analysis of clonal responses to treatment and revealed several interesting findings (Fig. 4f–i and Supplementary Fig. 18a–d). First, chemotherapy can affect genetically distinct clones differently such as in P69, an infant ALL that relapsed in maintenance 1 during treatment with 6-MP, with an *NT5C2*[R39Q] in 28% of the relapse cells, where a change to the ALLR3 protocol without thiopurines eradicated the *NT5C2*-containing clone. Second, a diagnostic clone that initially had the most rapid response to chemotherapy, reappeared to cause relapse after 147 days in remission (P11). Third, the relapse clone can be dormant for a long time before causing relapse, such as in P17, where it was undetectable for 378 days until it reappeared in 1% of cells, almost 100 days before expanding in a clonal sweep. To gain further insight into the clonal composition and mutational co-occurrence in this patient, single cells were sorted and selected targets were assessed (Fig. 4j, Supplementary Fig. 18e–k and

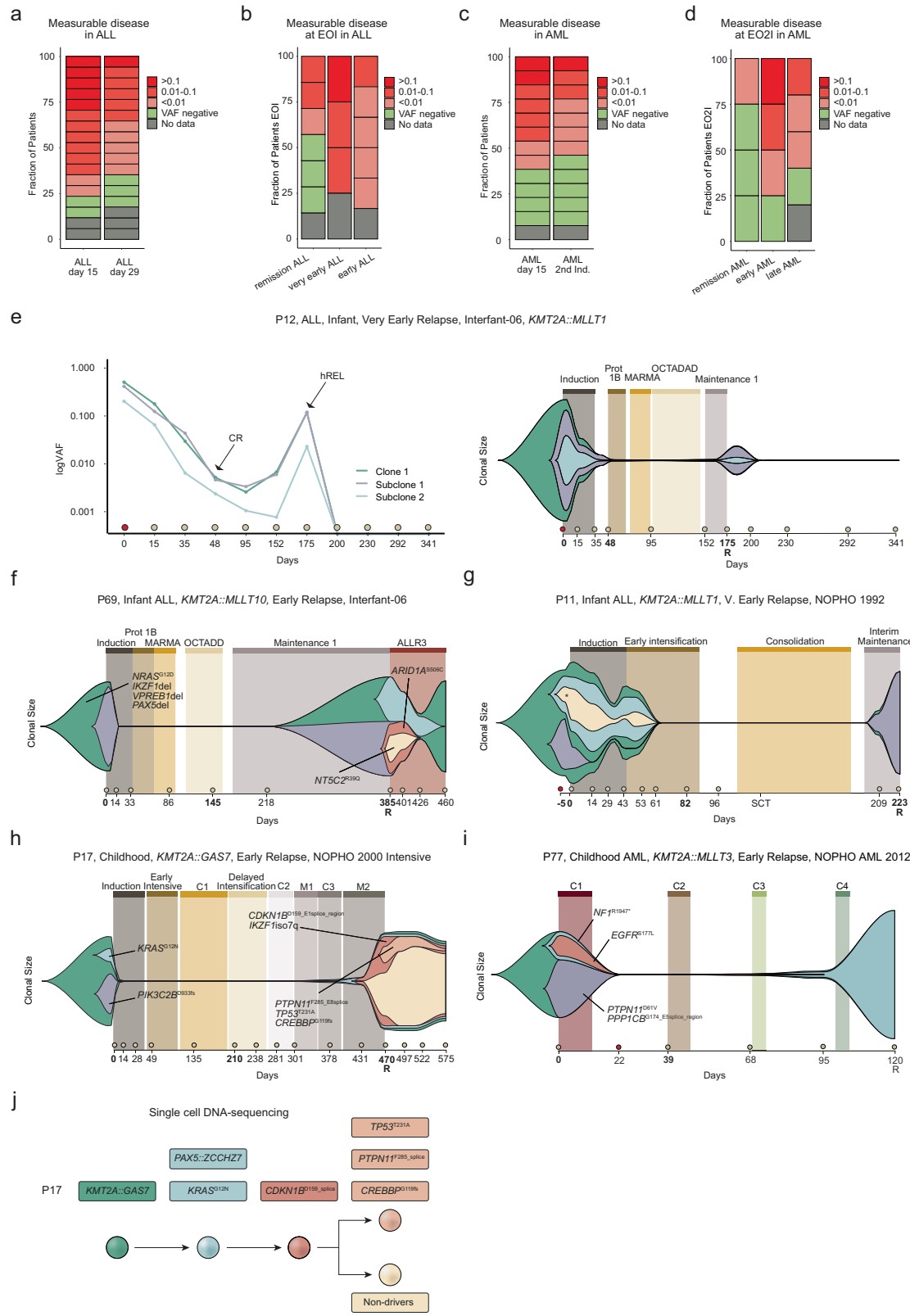

Supplementary Data 15). This showed that mutations in *CDKN1B* and
*KRAS* co-occurred in the dominant clone, with two subclones, one with
mutations in *TP53*, *PTPN11* and *CREBBP* in 40% of cells, and one without
additional mutations in the enriched pathways (20% of cells). We
conducted a similar analysis for P57 and verified clonal expansion of a
population exhibiting two *TP53*-mutations (Supplementary Fig. 12d).
Finally, different signaling mutations likely have distinct molecular

consequences as a minor diagnostic *NF1*-containing clone out-
competed a larger *PTPN11*-containing clone at relapse (P77). These
data demonstrate unique clonal responses to the treatment given.

## Discussion

We here demonstrate different mechanisms of relapse in *KMT2A*-r ALL
and AML. In ALL, early relapse is likely driven by surviving cells that

**Fig. 4 | Longitudinal analyses and clonal dynamics during treatment.**
**a–d** Leukemic burden (**a**) in ALL, days 15 and 29. **b** at the end-of-induction (EOI) in ALL (**c**), in AML, day 15 and after EO2I. **d** after EO2I for AML. **e** Linear diagram and fish plot showing clonal dynamics in P12. CR indicates complete remission by Fluorescence In Situ Hybridization (FISH) and hREL indicates hematologic relapse. Mutations were grouped into three clones and at CR, 1% of leukemia cells were detected. **f–i** Clonal evolution by fish plots, the treatment blocks are indicated and at the bottom, the day from diagnosis with circles, bone marrow (BM, grey), and peripheral blood (PB, red). Samples studied by targeted sequencing, WGS, and WES in bold, R=relapse. **f** An infant ALL that relapsed during maintenance. The major diagnostic clone contained deletions of *PAX5, IKZF1, VPREB1* and an *NRAS*^G12D (green) and gained an *ARID1A*^S506C (red) and an *NT5C2*^R39Q in 27% of cells (yellow) at relapse, ALLR3 was started (without thiopurines), and the *NTC52*^R39Q-clone became

undetectable. **g** For this infant, a PB-sample was taken 5 d before diagnosis; the purple clone became undetectable by day 19, and the leukemia cells increased to 60% 7 days later, CR was reached (by FISH), and the patient was transplanted. Around 3 months later, the diagnostic subclone (purple) that responded to induction therapy was detected (14%), and reached clonal dominance with no additional driver mutations. **h** A child where the *KMT2A*-r was detected at day 378 in 3% of cells, 92 days before relapse and 350 days after CR. The *KRAS*^G12N-relapse clone was detected at diagnosis (2%). **i** An AML where a subclonal *NF1*^R1947* outcompetes a larger *PTPN11*^D61V-clone at relapse. **j** The order of mutations at relapse as determined by single-cell sequencing. Mutations are gained sequentially with all relapse cells containing *KRAS*^G12N, *PAXS::ZCCHZ7*, *CDKN1B*^D159_EIsplice_region, followed by expansion of cells with *TP53, PTPN11*, and *CREBBP*-mutations and a clone without additional known driver events.

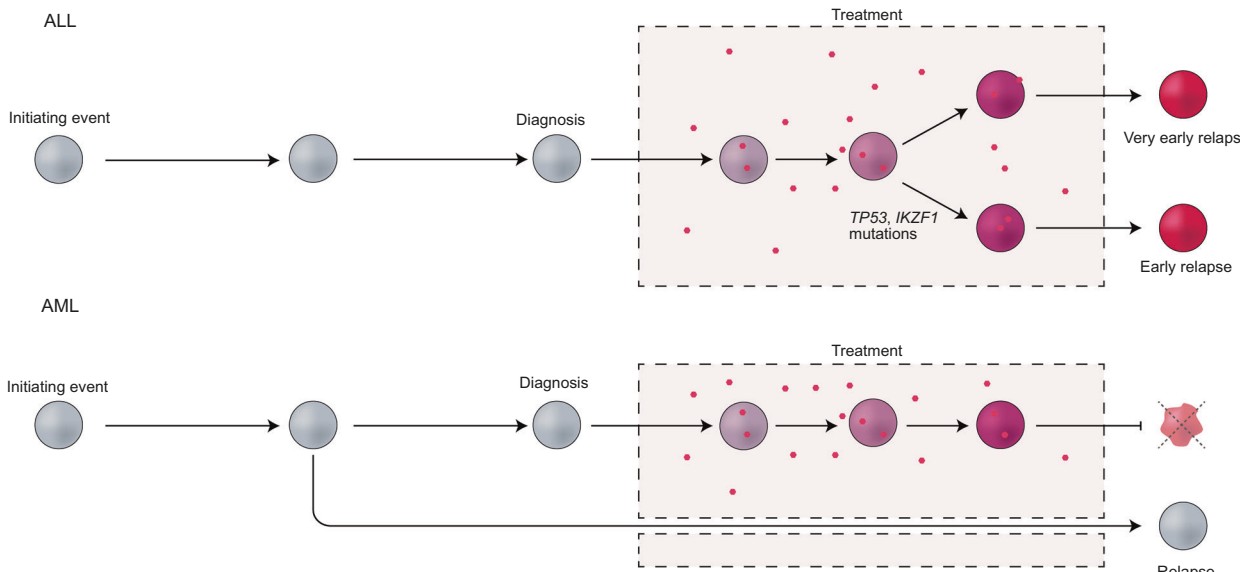

Relapse mechanisms in *KMT2A*-r infant and childhood leukemia

**Fig. 5 | Illustration of different relapse mechanisms in *KMT2A*-r leukemia cells.** Above, the cells in very early relapse ALL had a marked chemotherapy-exposure signature suggesting that the surviving cells accumulate mutations due to cyto-toxic cell damage, with 79% gaining mutations in drug-response genes, most commonly (69%) in *TP53* and *IKZF1*. Very early relapse ALL ( < 9 months) had a paucity of such alterations and lacked a prominent chemotherapy-exposure

signature, and 80% arose through multiple clones seeding relapse, suggesting inherent resistance. AML relapse (below) also lacked a marked chemotherapy-induced signature, instead the same mutational processes were active at diagnosis and relapse. This is in line with relapse from an evolutionary early cell that is dormant during therapy and therefore remains unaffected by therapy.

accumulate mutations because of chemotherapy exposure and in 79%, mutations in drug-response genes were detected at relapse, most commonly in *TP53* and *IKZF1*. By contrast, 91% of very early relapse ALL lacked such mutations, in line with inherent resistance. Relapsed AML had the same mutational processes active at diagnosis and relapse and lacked a chemotherapy-exposure signature, suggesting relapse from a cell that has remained unaffected by therapy (Fig. 5).

Multiple diagnostic clones seeding relapse were seen in 55% of patients overall and enriched in very early relapse/resistant *KMT2A*-r ALL (80% versus 33% in early relapse). This evolutionary pattern is seen in 20% of relapsed childhood ALL across genetic subtypes and thus enriched in *KMT2A*-r ALL[11]. Together with the general lack of new driver mutations, this implies that very early *KMT2A*-r infant ALL relapse is driven by intrinsic resistance where other factors, including the host genetics, microenvironment and/or cell state drive relapse[18]. Early relapse *KMT2A*-r ALL commonly evolved through a clonal sweep (67%) and harbored alterations in chemoresistance-associated genes including *TP53, IKZF1, NT5C2, NR3C1, WHSC1*, and *CREBBP* in 79%. Similar alterations are detected in 17% of very early relapse and 65% of early relapse ALL across genetic subtypes in pediatric ALL

(*n* = 94 non-*KMT2A*-r, *n* = 9 *KMT2A*-r) and restricting the analyses to non-*KMT2A*-r cases, in 22% and 70%, respectively[11]. Importantly, co-existing *IKZF1* and *TP53* alterations at relapse were not seen in any of these cases[11], suggesting that this may be unique to *KMT2A*-r ALL. *IKZF1* and *TP53* associate with inferior prognosis in childhood ALL[19–21] and recently, diagnostic *TP53* and *IKZF1* alterations were identified in 14% and 8% of adult *KMT2A*-r ALL respectively, and found to be associated with poor outcome[22]. These alterations are rare at diagnosis in infant and childhood *KMT2A*-r leukemia (4% and 0%, respectively) and highly enriched at relapse herein (64%)[7]. To validate our findings, we analyzed data from 18 *KMT2A*-r ALL trios[7,8,11], showing a frequency of chemotherapy resistance-associated alterations on a par with our data in early relapse cases (50%, 6/12), and a paucity of such changes in very early relapse cases (0/6). Still, the number of patients analyzed remains low, and some lacked complete genomics data, thus larger studies are needed to confirm our results. In agreement with the patient data, functional studies have demonstrated that alterations in *TP53* and *IKZF1* drive resistance to numerous cytostatic, suggesting that alternative treatments are needed to successfully treat these patients[11,23–29].

In line with the possible acquisition of *TP53*-alterations during treatment[11,25,30], they were not detected at diagnosis with targeted sequencing including duplex-sequencing in P58[30]. Similarly, remaining drug-associated mutations were gained during maintenance therapy with thiopurines and/or anthracyclines, and not detected at diagnosis[9,10,31]. This is, to the best of our knowledge, the first study demonstrating that *IKZF1* and *TP53* are tightly connected to early relapse in *KMT2A*-r ALL, detected in 64% of cases, suggesting monitoring during treatment for the rise of these high-risk lesions may allow for early intervention(s).

Early relapse AML maintained most mutations within the enriched pathways while later relapses had a higher mutational turnover. Diagnostic *TP53* or *CCND3* alterations were detected in 40% of early relapse AML and may mark patients with a high relapse risk, warranting future studies. *CCND3* mutations are rare in *KMT2A*-r AML (8.9%)[32], and enriched in relapse cases (16.7% herein). Recurrent 12p deletions affecting *ETV6* and *CDKN1B* were identified at relapse, and as the minimally deleted region does not always contain *ETV6*, *CDKN1B* may be the target gene[15]. 12p deletions associate with inferior survival in childhood AML, are often cytogenetically cryptic/complex and may lead to *ETV6*-fusions[33]. Also our cases had complex karyotypes and did not survive, and although no *ETV6*-fusions were identified, a *SETD2::DCP1B*-fusion was detected which caused loss of *SETD2*. Only one *SETD2*-fusion has been described in leukemia, a *SETD2::CCDC12*, that we identified in a child with *KMT2A*-r ALL[7]. Finally, while RAS-signaling mutations were maintained at early relapse, they were lost at late relapse, which instead gained kinase receptor mutations, suggesting variable dependence on signaling pathways.

The mutational signature analysis revealed a difference in the mechanisms of relapse in *KMT2A*-r ALL and AML and between very early and early relapse ALL. Relapsed ALL had a large number of new mutations and in early relapse ALL, most mutations gained at relapse had a chemotherapy-exposure signature and likely occurred due to treatment, as also suggested by others[11,34]. Thus, surviving cells accumulate mutations during treatment until a point where they expand to cause relapse, often accompanied by mutations in drug-response genes (79%)[11,25,31]. By contrast, the chemotherapy-exposure signature was not prominent in very early relapse infant ALL, and their short time to relapse rather points towards an inherent resistance[18]. Interestingly, the primary signature in the three refractory ALL cases was of unknown origin, indicating a potential common yet unidentified cause. In relapsed AML, the pattern of mutations was similar at diagnosis and relapse and mirrored the natural acquisition of mutations, rather than being caused by chemotherapy (two infants and seven children were analyzed). This demonstrates the absence of additional mutational pressure and that an evolutionary early and inherently resistant or dormant cell, which remained unaffected by treatment, caused the relapse. This finding is consistent with recent single-cell RNA-sequencing data showing a shift at relapse toward a more primitive cellular state in *KMT2A*-r AML[35]. Although the strong chemotherapy-induced signature in ALL and the more diagnosis-like signature in AML have been seen previously, this difference has not been highlighted before[11,36]. These different modes of resistance are important as adjusted doses might be required for persistent cells that acquire mutations along treatment, whereas for inherently resistant cells, new treatments are likely needed, and dormant cells might be targeted by being pushed into cell division. Combined, new therapies including Blinatumomab, CAR-T, or Menin inhibitors, are likely needed to conquer all types of relapsed disease.

Few have analyzed childhood acute leukemia during treatment with NGS[11,37]. Our data extend these studies to include infant leukemia and showed that a change in therapy can favor the eradication of one clone and expansion of another demonstrating variable sensitivity to the given drugs. Further, a diagnostic clone that initially was the most sensitive clone to therapy can reappear to cause relapse, suggesting

awakening by changes in the microenvironment, signaling, immune system or a combination thereof[38]. Mutations in the same pathway likely provide different cellular advantages and can outcompete each other at relapse and finally, a stepwise replacement of a fitter clone was seen in cases with multiple relapses.

Given the prognostic value of MRD in childhood ALL[39,40], we investigated measurable disease using personal primers, and very early relapse ALL still had high levels of leukemia cells at the EOI, whereas cases that stayed in remission lacked detectable leukemia cells. In AML, the fraction of leukemia cells after the second induction has prognostic significance[41] and herein 6/7 cases with measurable disease at that time relapsed. All but two cases were in CR by FC, highlighting the sensitivity of NGS[42,43]. However, the limited cohort size may affect the results, underscoring the need for validation in larger studies. Molecular monitoring with personal mutations detected relapse up to 3 months before clinical relapse and could be an important clinical tool as earlier detection increases the treatment window and chances of survival[44].

Collectively, this study provides unique insights into the mechanisms of relapse in a highly aggressive leukemia. Mutations in chemoresistance-associated genes were identified in 79% of early relapse ALL with co-existing *TP53* and *IKZF1*-alterations being highly enriched (64%) while almost all very early relapse ALL lacked such alterations. Our results suggest that some of the mutations that drive early *KMT2A*-r relapse ALL accumulate in persisting leukemia cells because of chemotherapy treatment, while very early relapse ALL is driven by an inherent resistance. In contrast to ALL, *KMT2A*-r relapse AML stems from an evolutionary early and likely dormant cell that remains unaffected by chemotherapy exposure. These results have implications for future treatment strategies and prediction of relapse.

## Methods

### Patient cohort

Informed consent was obtained according to the Declaration of Helsinki and the study was approved by the local Ethics Committee of Lund University, Sweden. The serial samples were made possible by a practice of periodical bone marrow monitoring during therapy. The cohort consisted of 52 *KMT2A*-r patients diagnosed between 1992 and 2010 and treated on the Nordic Society of Paediatric Haematology and Oncology (NOPHO), Interfant protocols, AALL0631, or TINI (Supplementary Data 1). The median time from diagnosis to relapse was 405 days for infant ALL (range 60–1024), 419 days for childhood ALL (range 378–459), 372 days for infant AML (range 60–683) and 205 days for childhood AML (range 104–337). The ALL cohort was divided into those that stayed in remission ($n = 7$), very early relapse/resistant disease if relapse occurred <9 months after diagnosis or if they never reached CR ($n = 11$) and early relapse if relapse occurred 9–36 months from diagnosis ($n = 16$, note that for P22 and P56, the paired relapse sample was not available as they occurred in the testis and CNS, respectively), based on the definition by Li et al.[11]. The AML cohort was divided into those that stayed in remission ($n = 4$), early or late relapse if relapse occurred before ($n = 6$) or after 1 year in CR ($n = 8$), based on the clinical definition[45].

### DNA extraction

DNA was extracted from 267 BM and 45 peripheral blood (PB) samples as a part of clinical procedures ($n = 119$), from TRIZOL ($n = 42$), fixative ($n = 23$), cell pellets ($n = 87$), or glass slides ($n = 40$) using standard protocols including TRIzoITM Reagent (Thermo Fisher Scientific, Waltham, MA, USA), Gentra Puregene Blood Kit (Qiagen, Hilden, Germany), QIAamp DNA Micro kit (Qiagen) or an in-house protocol for fixative (Supplementary Data 12). From cells in TRIzolReagent (Thermo Fisher Scientific), DNA was isolated from the interphase with an in-house protocol. In brief, 500 µl Back extraction buffer (50 mM Sodium Citrate (Sigma-Aldrich, St Louis, MO), 1 M Tris (Thermo Fisher

Scientific) and 4 M Guanidintiocyanat (Sigma-Aldrich) were added to the interphase, followed by 55 °C for 5 min, 10 min mixing and 12,000 x g, 30 min. The water phase was mixed with 1 µl Glycogen (10 µg/µl, Invitrogen, Waltham, MA, USA) and 800 µl 2-propanol (Sigma-Aldrich), incubated for 5 min, followed by 12,000 x g, 15 min at 4 °C. The pellet was washed with 80% Ethanol (Solveco Rosersberg, Sweden) and resuspended in nuclease-free water (Invitrogen). The phase lock gel system (Light, QuantaBio, Beverly, MA, USA) was used for clean-up after extraction.

For samples in fixative, cells in 200 µl freshly made fixative were neutralized with 140 µl 5 M NaOH (Sigma-Aldrich), lysed for 10 min at RT with 0.32 M Sucrose (Merck), 10 mM Tris (VWR, PA, USA), 5 mM $MgCl_2$ (Sigma-Aldrich), and 1% Triton X-100 (Sigma-Aldrich), followed by 350 x g, 5 min, resuspension in 50 µl Extraction buffer (1 M Tris (pH 8), 0.5 M EDTA (Sigma-Aldrich), 20% SDS and Proteinase K (0.2 µg/µl, Qiagen) and 1 h at 60 °C, followed by clean-up using the phase lock gel system.

Slides were incubated with 100% xylene (Sigma-Aldrich) until the cover slide detached, rinsed with Ethanol (70, 85, 99.5%) and then milli-Q water. The cell smears were scraped off into 180 µl ATL-buffer and 30 µl proteinase-K and extracted using the QIAamp DNA Micro kit (Qiagen). DNA quantity and quality were assessed by the Qubit 4 fluorometer and NanoDrop One (both Thermo Fisher Scientific), respectively.

## Whole-genome, whole-exome sequencing and bioinformatic analysis

Whole-genome libraries were performed using the TruSeq Nano protocol (Illumina Inc, San Diego, CA, USA) and 150 bp paired-end sequencing was performed using Illumina HiSeqX or NovaSeq 6000. Whole-exome libraries were performed using the Nextera Rapid Capture Exome kit (Illumina) (Nextera rapid capture exome 15037436 Rev.D) or Twist Comprehensive Exome (Twist Bioscience, San Francisco, CA, USA), and 300 bp paired-end sequencing was performed on Illumina NextSeq500 or NovaSeq6000.

Methods used for mapping, coverage and quality assessment, SNV and Indel-analysis, tier annotation, and Loss of Heterozygosity (LOH) detection have been described in ref. 17. In short, for the annotation of transcripts, Ensembl (build 54_36) and Genbank (build downloaded on 21 May 2009) were used. The following four tiers were used to classify sequence variants (i) tier 1: coding synonymous, non-synonymous, splice-site and noncoding RNA variants; (ii) tier 2: conserved variants (conservation score cutoff of greater than or equal to 500, based on either the phastConsElements28way table or the phastConsElements17way table from the UCSC Genome Browser) and variants in regulatory regions annotated by UCSC (regulatory annotations included are targetScanS, ORegAnno, tfbsConsSites, vistaEnhancers, eponine, firstEF, L1 TAF1 Valid, Poly(A), switchDbTss, encodeUViennaRnaz, laminB1 and cpgIslandExt); (iii) tier 3: variants in non-repeat masked regions; and (iv) tier 4: remaining SNVs. For structural variations, CREST[46] was used and for CNA, CONSERTING[47]. Both SVs and CNAs were manually inspected. Zoomed-in allelic imbalance plots were generated for *PAX5, IKZF1*, and *CDKN2A/B*, to ensure that focal CNAs were not missed.

A subset of samples was analyzed as follows (St Jude, CAB pipeline); Paired-end reads were aligned to human reference genome build hg38 using BWA-mem that is implemented in the tool 'fq2bam' from NVIDIA Clara Parabricks GPU accelerated toolkit. For tumor/normal variant calling, an ensemble strategy with optimized filtering (SNV called by at least 2 callers) was employed to identify SNVs/indels using Mutect2 (v4.1.2.0)[48], SomaticSniper (v1.0.5.0)[49], VarScan2 (v2.4.3)[50], MuSE (v1.0rc)[51], and Strelka2 (v2.9.10)[52]. Variant annotation was performed using Annovar[53]. These consensus calls were further undergoing manual assessment and evaluation of read depth, mapping quality, and strand bias, to eliminate additional artifacts. Tumor/normal matched samples were further subjected to copy number variant (CNV) calling using CONSERTING[47] and CNVKit[54], as well as SV calling by Gridss[55], Manta[56], and DELLY[57]. The results were assembled and QCed to keep high-quality SVs. For P30, SNP-array data was included for the diagnostic sample[58]. P137 and 138 were analyzed with Dragen.

P133 was analyzed using another pipeline[36]. Briefly, pair-end reads were aligned to hg38 with BWA[59]. Duplicate reads were marked with Picard and Indel realignment was performed with GATK[48]. Somatic SNVs and indels were called using MuTect, MuTect2 and MuSE[48,51]. Variants were included if either 1) Identified in both WGS and WES by at least two out of three variant callers, or 2) Identified by either WGS or WES by all three variant callers (MuSE, Mutect, and Mutect2). SVs were identified by Manta[56], DELLY[57], novoBreak[60], and SvABA[61], all with default settings. Patchwork was used for CNV-calling[62]. Mutations were pre-filtered based on sequencing depth in tumor and normal sample, sequencing reads for the mutant allele, and allele frequency in the population.

Patients lacking a germline sample were analyzed against a random germline and only genes that were coupled to relapse herein, or in published cohorts[11,37] were studied. Putative SNVs were excluded if they had a normal allele frequency >0.01 (Exome Aggregation Consortium, reference variants (rs) and SWEfreq if the germline VAF was >0.2, and if mutant reads were only on one strand. For structural variants only known alterations were included.

## Mutational signatures

The R-package MutationalPatterns[63] 3.14.0 was utilized to assess the relative contribution of single base substitutions in non-repetitive regions identified through WGS to the Cosmic SBS mutational profiles. Each signature was classified as Clock-like, APOBEC/AID, Mismatch Repair (MMR), UV, Chemotherapy treatment, Biological, ROS, Tobacco, Chemical exposure or Unknown based on their description in Cosmic, and each patient was assigned to a "primary signature" based on their most common classification (Supplementary Data 11)[16]. When calculating the average VAF per signature, each mutation was first assigned to the most likely signature: All 96 possible motifs were designated a most likely signature, by mapping each one to the Cosmic SBS mutational signature based on where the motif has the highest contribution.

## Pathway analysis and relapse-specific mutations

Pathways were included if they contained >1 relapse-specific alteration in paired samples. Glucocorticoid receptor signaling was included as it is connected to resistance. All mutations were evaluated using DriverPower[64] and Annovar[53]. The heatmap and location of relapse mutations were generated using St Jude's webtool ProteinPaint (https://pecan.stjude.cloud/proteinpaint/study/). For CNAs, focal CNAs (≤5 genes), broad CNAs (>5 affected genes but less than 400), and whole-arm changes were included if detected at relapse or if in a recurrently mutated pathway. The genomic random interval (GRIN) analysis was used to identify recurrently mutated genes[65]. Whole chromosome gains were excluded unless there was also a focal deletion/amplification, and CNAs connected to *KMT2A*-rearrangements were excluded. Alterations in unpaired samples were included if they were present in a recurrent pathway and were known cancer-associated mutations.

## 2D plots

SciClone[66] was used to determine clonal evolution patterns. Validated tier 1 and high-quality tier 2 and 3 SNVs were included together with manually inspected CNAs. If the variant had been studied by targeted sequencing, the VAF with the highest coverage was used and relapse variants detected in a low fraction of diagnostic cells in the targeted sequencing were also included. The minimum depth for a variant was 10 reads.

## Primary Template-directed Amplification (PTA)

For 13 samples with no measurable or limited amounts of DNA (<0.1–3.2 ng), the "ResolveDNA Whole Genome Amplification v.1 kit" (Bioskryb Genomics, Durham, UK) was used for whole genome amplification (WGA) (Supplementary Data 12). PTA was performed according to the manufacturer's instructions with modifications. The lysis step was performed for 12.5 min, 1400 rpm at RT (MPS-1 plate mixer, BioSan, Riga, Latvia). During clean-up, the BioSkryb ResolveDNA beads and WGA-DNA were resuspended by mixing for 2 min, 1500 rpm at RT (MPS-1, Biosan) followed by 1 min, 19xg at RT. After ethanol wash, resolved WGA-DNA was resuspended by mixing for 2 min, 1700 rpm at RT and 1 min, 19xg at RT. WGA-DNA was quantified with a dsDNA high sensitivity kit using Qubit 4 (Thermo Fisher Scientific) and fragment size determined with D5000 ScreenTape assay using the 4200 TapeStation (Agilent Technologies, Santa Clara, CA, USA). A non-targeting control and 100 pg genomic DNA were used as a negative and positive control, respectively.

## Longitudinal deep-sequencing

Between 6 and 35 tier 1-3 mutations from WGS/WES data were selected per patient. Primers were designed using Primer3[67] and PrimerTK (https://github.com/stjude/PrimerTK), where costume scripts were used to check for dimer formation and an In-Silico PCR tool for off-target products and ordered from Integrated DNA Technologies (IDT, Coralville, Iowa, USA) (Supplementary Data 16). PCR was performed using Qiagen Multiplex PCR Kit (Qiagen) with 5-20 ng input DNA and products were quantified using Qubit 4 (Thermo Fisher Scientific) and prepared for sequencing using Nextera XT DNA Sample Preparation Kit, Index Kits (Illumina), and purified using AMPure XP beads (Beckman Coulter Inc., Brea, CA). 2×150 bp paired-end sequencing was performed using Illumina NextSeq 500, MiniSeq or MiSeq. Reads were trimmed using Trimmomatic (0.38)[68] and paired-end reads were aligned to hg19 using BWA (0.7.15)[59]. PCR duplicates were marked using Picard (2.6.0, Broad Institute, Cambridge, MA, USA). Variant calling was performed using Freebayes[69] (1.1.0) (parameters: --min-alternate-fraction 0, --strand-filter 0, --p-value 1, --min-mapping-quality 50).

## Dilution series to determine the sensitivity of the multiplex-PCR

Kasumi-1 (ACC-220, DSMZ, Braunschweig, Germany), carrying a homozygous TP53 (chr17:7577538, C > T) mutation was diluted with REH (ACC-22, DSMZ), lacking TP53 mutations. Cells were cultured under standard conditions and DNA was extracted using the DNeasy Blood & Tissue kit (Qiagen) and diluted to 11 ng/µl. Kasumi was diluted with REH in nine steps, mixed and incubated for 15 min before the next dilution to an expected VAF of 0.0019. This included up to four multiplex PCR replicas. Two additional TP53-mutations were found in Kasumi-1 (chr17:7577407, A > C and chr17:7577427, G > A).

## Single-cell sorting and whole genome amplification

Relapse cells from P17 and P57 were thawed in 10% FBS (HyClone, Logan, UT, USA) with DNase I (1 mg/ml, Roche, Basel, Switzerland) and RPMI1640 (Thermo Fisher Scientific), 1% Penicillin-Streptomycin (Thermo Fisher Scientific), centrifuged for 5 min, 350 x g at 4 °C and resuspended in PBS + 2% FBS (HyClone). Cells were blocked with Human TruStain FcX (BioLegend, San Diego, CA, USA), according to the manufacturer's description, and stained for 30 min at 4 °C with antibodies and isotype controls (Supplementary Data 17). Cells were washed and resuspended in PBS + 2% (HyClone) and incubated with 7-AAD (BD Biosciences, Franklin Lakes, NJ, USA) for 10 min at 4 °C before flow analysis.

Single cells were isolated by FACS Aria Fusion (BD Biosciences) with FACSDiva v8.0.2 (BD) using index sorting. The precision was set according to "A Rapid Method to Verify Single-Cell Deposition Setup for Cell Sorters"[70]. Cells were sorted into twin.tec LoBind PCR plates (Eppendorf) with 1.5 µl Cell Buffer (ResolveDNAv.1, BioSkryb), using

the 100 µm nozzle, covered with plastic film (Thermo Fisher Scientific), spun down (MPC-25), mixed for 15 s, 2300 rpm (MPS-1, BioSan, Riga, Latvia), spun down (MPC-25) put on dry ice and stored at −80 °C. Whole genome amplification, PCR and library preparation were performed as above (Supplementary Data 16). Libraries were sequenced using MiSeq (Illumina).

## Clonal evolution visualized by fish plots

Fish plots were generated using the "fishplot" package in R"[71]. The clusters were determined manually based on the VAFs and some VAFs were adjusted to draw the fishplots. If available, single-cell data were used for interpretation. If samples were missing close to or after the relapse in patients with multiple relapses, we have depicted the relapse clone as if it emerges and disappears 20 days before and/or after the relapse and 30-50 days before the single/last relapse for illustrative reasons. The VAFs for SNVs in regions with CNA were corrected.

## Variant allele frequencies at the MRD time points

The VAF for a given timepoint was an average of the two highest VAFs and we required that the KMT2A-r was identified. However, given that patients often are cytopenic at early time points and that the number of leukemic cells is low/undetectable during remission, we rescued the sample despite the KMT2A-r was not identified (`n = 8`) if >2 variants from the major diagnostic clone and/or relapse had a VAF above our detection level (>0.002). We required that no other mutations were present at the site with a VAF > 0.002 indicating artifacts.

## CRISPR/Cas9 genome editing of PRPS2

The human REH (CRL-8286) B-ALL cell line was purchased from the American Type Culture Collection (ATCC) and authenticated by STR analysis. Cells were cultured in RPMI 1640 (Gibco, 11875093) supplemented with 10% fetal bovine serum (FBS, HyClone, SH30071.03) at 37 °C with 5% $CO_2$. PRPS2 knockout single clones were generated using CRISPR-Cas9 and were confirmed by Next-Generation Sequencing (See Supplementary Data 18 for sgRNA sequence).

## Plasmid construction and lentivirus production

PRPS2 (NM_001039091) wildtype and PRPS2[P320L] were synthesized and subcloned to MSCV-IRES-GFP using EcoRI cloning site using Genescript (Piscataway, NJ, USA). Human PRPS2 wildtype or mutant PRPS2 P320L were amplified by PCR and cloned into the cl20-Elongation Factor 1 alpha (EF-1α)-internal ribosome entry site (IRES)-GFP lentiviral plasmid using NEBuilder HiFi DNA Assembly Master Mix (NEB, E2621L)[72]. Purified cl20c-IRES-GFP empty vector, cl20c-Flag-PRPS2-WT-IRES-GFP, and cl20c-Flag-PRPS2-P320L-IRES-GFP were transfected with packaging vectors into Lenti-X 293 T cells (Takara, #632180)[73,74]. After 48 h, the culture media containing lentiviral particles were collected to infect REH (CRL-8286, ATCC) parental and PRPS2 KO cell lines for another 48 h. The transduction efficiencies were determined by flow cytometry. Primers used for plasmid construction can be found in Supplementary Data 18.

## Western blotting

After being washed once with cold PBS buffer, the cells were lysed by RIPA lysis buffer (Thermo Scientific, #89901). Protein concentrations were determined using BCA assay kit (Thermo Scientific, #23228). 30 µg protein for each sample was subjected to western blotting. Flag-tag antibody (Invitrogen, #PA1-984B-HRP, Lot No. WC316048, 1:1000 dilution) was used for the detection of overexpressed PRPS2. Vinculin level for each sample was detected with a vinculin antibody (CST, #13901, Lot No.7, 1:1000 dilution).

## Drug sensitivity assessment

CellTiter-Glo (CTG) assay (Promega, #G7573) was used to test cell viability. REH-cells were seeded into a 384-well microplate (Thermo

Scientific, #242765) with 2,000 cells/well. Gradient-diluted 6-MP was then added and incubated for 72 h until the CTG-assay was performed and tested by microreader (Agilent, Biotek, Synergy H4 hybrid reader). The assay was repeated twice.

## Statistics

R 4.4.0 was used for statistical calculations. Spearman correlation test was used for comparison of signature analysis at diagnosis and relapse in AML, as the data was significantly different from a normal distribution, determined by Shapiro Wilk normality test. The comparison of the number of gained mutations across *KMT2A*-r was tested by Kruskal-Wallis. For all two group comparisons, a 2-sided Wilcoxon rank sum test was used. For Kaplan-Meier survival estimations, Survival 3.5.8 was used.

## Reporting summary

Further information on research design is available in the Nature Portfolio Reporting Summary linked to this article.

## Data availability

The sequencing data generated in this study is deposited in the European Genome-phenome Archive (EGA). The sequencing data are available under restricted access as it is considered personal data and falls under the General Data Protection Regulation (GDPR), and access can be obtained by upon request from the corresponding author (Anna Hagström-Andersson, anna.hagstrom@med.lu.se) through EGAS00001008197. The raw sequencing data are protected and are not available due to data privacy laws. Source data are provided with this paper and the sequencing data generated in this study are provided in the Supplementary Information/Source Data file. Source data are provided with this paper.

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

## Acknowledgements

This research was supported by The Swedish Childhood Cancer Fund (20121-0046, AKHA), The Swedish Cancer Society (20-1036, AKHA), The Swedish Research Council (2019-01446, AKHA), The Knut and Alice Wallenberg Foundation (2014-0098, AKHA), The Crafoord Foundation (2726-001, AKHA), The Nilsson-Ehle Donations, Ellen Bachrach Memorial Fund 2022-04, Governmental Funding of Clinical Research within the National Health Service, all AKHA. Sequencing was performed either by the SNP&SEQ Technology Platform, Uppsala, the National Genomics Infrastructure (NGI) Sweden and Science for Life Laboratory or at The Center for Translational Genomics, Lund University and Clinical Genomics Lund, SciLifeLab. The SNP&SEQ Platform is also supported by the

Swedish Research Council and the Knut and Alice Wallenberg Foundation. We thank the Center for Advanced Genome Engineering (S. Miller, A. Loughran and T. Caera) for their technical support in performing experiments included in this study. We thank Elias Levy Itshak Salfati and Pratima Nallagatla for bioinformatics support.

## Author contributions

L.A. and A.K.H.A designed the study; L.A., H.S., and V.S. performed experiments; M.M., H. Z., and J.J.Y. performed functional assays, K.L.P. performed structural modeling of *PRSP2*; L.A., M.P., M.Y., M.P.W., G.S., J.M., Z.C., and J.Z. performed computational data analyses; L.A., M.P and A.K.H.A analyzed sequencing data. A.C., C.J.P., P.S., G.B., K.P.T., J.A., L.F., H.V.M., B.L., R.P., T.A.G., R.W.S., P.M. and O.L. collected patient material and clinical data; L.A., M.P., and A.K.H.A wrote the manuscript, and all other authors performed critical reading of the manuscript.

## Funding

## Competing interests

The authors declare no competing interests.

## Additional information

[1]Division of Clinical Genetics, Department of Laboratory Medicine, Lund University, Lund, Sweden. [2]Department of Pathology, St. Jude Children's Research Hospital, Memphis, TN, USA. [3]Department of Pharmacy and Pharmaceutical Sciences, St Jude Children's Research Hospital, Memphis, TN, USA. [4]Center for Applied Bioinformatics, St Jude Children's Research Hospital, Memphis, TN, USA. [5]Childhood Cancer Center, Skåne University Hospital, Lund, Sweden. [6]Department of Clinical Immunology, National University Hospital, Rigshospitalet, Copenhagen, Denmark. [7]Department of Clinical Medicine, Faculty of Health and Medical Sciences, University of Copenhagen, Copenhagen, Denmark. [8]Department of Paediatrics and Adolescent Medicine, Rigshospitalet, University of Copenhagen, Copenhagen, Denmark. [9]Princess Máxima Center for Pediatric Oncology, Utrecht, The Netherlands. [10]Department of Molecular Medicine and Surgery, Karolinska Institutet, Stockholm, Sweden. [11]Department of Oncology and Pathology, Karolinska Institutet, Stockholm, Sweden. [12]Department of Pediatrics, Institution for Clinical Sciences, Sahlgrenska Academy, University of Gothenburg, Gothenburg, Sweden. [13]Tampere Center for Child, Adolescent and Maternal Health Research, Faculty of Medicine and Health Technology, Tampere University, and Tays Cancer Center, Tampere University Hospital, Tampere, Finland. [14]Region Västra Götaland, Sahlgrenska University Hospital, Department of Clinical Chemistry, Gothenburg, Sweden. [15]Department of Laboratory Medicine, Institute of Biomedicine, University of Gothenburg, Gothenburg, Sweden. [16]Josep Carreras Leukemia Research Institute and School of Medicine, University of Barcelona, Barcelona, Spain. [17]Institució Catalana de Recerca i Estudis Avançats (ICREA), Barcelona, Spain. [18]Department of Biomedicine, School of Medicine, University of Barcelona, Barcelona, Spain. [19]Spanish Cancer Network (CIBERONC), ISCIII, Madrid, Spain. [20]Pediatric Cancer Centre Barcelona-Sant Joan de Deu Hospital (PCCB-SJD), Barcelona, Spain. [21]Department of Computational Biology, St. Jude Children's Research Hospital, Memphis, TN, USA. [22]Department of Experimental Medical Science, Medical Structural Biology, Lund University, Lund, Sweden. [23]Department of Pediatrics, Stanford University School of Medicine, Stanford, CA, USA. [24]Center for Translational Genomics, Lund University, Lund, Sweden.
✉e-mail: Anna.Hagstrom@med.lu.se

