## [Transparent Peer review file · Nature Communications]

The genomic landscape of relapsed infant and childhood KMT2A-rearranged acute leukemia

Corresponding Author: Dr Anna Hagstrom-Andersson

Version 0:

Reviewer comments:

Reviewer #1

(Remarks to the Author)

This is a nice contribution into this rearranged subtype of ALL. Obviously it is a challenge to have a large number of patients, and the authors have done a nice job in presenting their findings, in particular the contrast between mutational signatures in relapse between very early and early patients with ALL, as well as the contrast in changes between diagnosis and relapse in patients with AML. The paper is extremely well written and the investigation is very thorough (especially the additional single cell analysis in some patients to tease out the temporal development of mutations in the longitudinal analyses).

Only a couple of points that could make the paper completely ready for publication.

1) The patients were accumulated over 18 years, and it should be more clearly stated which samples had bioinformatic analyses performed simultaneously on stored samples and which were performed in near real-time as the sample was obtained. This is important as a multitude of changes can occur over time if samples were not analyzed for mutations simultaneously - from changes in technicians, to changes in software, target sequencing depths, etc. - all of which can affect the ability to detect mutations. The largest finding of 75% vs 0% having TP53 or IKZF1 mutations should be very clear in this respect. If samples were not analyzed under identical circumstances - then efforts to show that such a finding was not due to such changes should be made. E.g., not all samples had a germline sample for a paired tumor-normal analysis. To alleviate concerns that lack of a germline sample in some patients could be influencing the results, an ad-hoc analysis in which only tumor samples from all patients are analyzed identically may show (hopefully) that the contrast between ages is not likely to be artifactual for this TP53/IKZF1 finding.

2) A rather large number of statistical comparisons are performed on this rather small number of patients. Hence, some subset of these findings are likely to be the result of multiple comparisons - and this should be stated as a limitation and that many of these findings likely need labeled as exploratory observations in need of further confirmation. The main hypothesis of the paper seems to have been the contrast in mutational signatures at relapse between very-early and early ALL patients having this rearrangement. Beyond that (or possibly also allowing for the contrast with AML), the other observations are definitely intriguing and hypothesis generating, but should likely all be grouped under this explorative caveat.

minor:

1) in the first paragraph, please provide a directly comparable contrast between general and infant KMT2A-r ALL (either both EFS or both OS).

2) Lines 51-53, please make it easier on the reader to tell how many with ALL and AML were infants and children (still can't tell from figure 1 easily).

3) Line 555 the single word Invitrogen needs removed

4) Somewhere in the introduction it may help the readership to clearly state the cohort selection criteria being based on KMT2A rearrangement at diagnosis as opposed to at relapse, and to indicate the percent of patients having detectable KMT2A rearrangements at relapse.

5) The bioinformatic section of the Methods there should be at least some summary statistics of the depths of sequencing obtained with the different techniques. Especially nice if that information was included specifically for depth of coverage of the drug-response genes - perhaps even with a supplemental figure showing that for all samples and regions of those genes

(at least for TP53 and IKZF1).

Reviewer #2

(Remarks to the Author)

In the current manuscript, the authors present a comprehensive exploratory NGS based genetic analysis of infant and childhood KMT2A-rearranged acute leukemia. By sequencing serial time-point samples of 33 ALL and AML patients with WGS, WES and concomitant deep sequencing they not only provide an important detailed description of the complex genetic and clonal landscape of this rare and aggressive subentity of ALL/AML, but also make some interesting novel discoveries. These are namely that very early versus early forms display differential mutational patterns, which suggest different biologic mechanisms of relapse, i.e. early ALL relapse is characterized by enriched acquisition of novel and frequently combined TP53 and IKZF1 mutations, while the analyzed cohort of very early ALL relapse is void of these mutations. Moreover, while relapsed ALL cases have an enrichment of chemotherapy-induced signatures, AML did not contain these signatures, also suggesting inherent differences of relapse between these two entities. The manuscript is well written and structured. The data has been thoroughly analyzed and interpreted and provides an important database for further research in these leukemia entities.

There are some major suggestions for improvement of the manuscript:

The strongest and most important discovery in this work are the differential mutational profiles between very early ALL relapse versus the early relapse cohort, which supports the hypothesis formulated by authors that early relapse is possibly driven by selection of cells with TP53 + IKZF1 mutations under the pressure of chemotherapy as opposed to an underlying primary / inherent resistance in very early relapse. However, the underlying data set is so far only descriptive and despite respectful acknowledgement of the collection of these rare primary samples, which has taken many years, the sample set (n=9 very early and 12 early, Fig. 1d) is still quite low to rank this as a strong conclusion but rather limits it to a strongly suggestive hypothesis. The manuscript would gain great improvement if the investigators were able to support this observation by some sort of further confirmation, which could e.g. either be by a targeted validation (i.e. not necessarily WES or WGS) of this observation in another cohort, or by functional validation of the hypothesis in corresponding pre-clinical models e.g. by showing that KMT2A-rearranged samples with (e.g. engineered) TP53 + IKZF1 mutations are more resistant to the applied therapies than original samples without these mutations. – Or if these preferable approaches are not feasible, at least an evaluation by a statistician of the data in Figure 1d and a critical discussion, that the evidence is limited by missing functional data and low sample numbers.

Minor:

line 44: language style: "...cells that gave rise to relapse was...": probably better if formulated: "...cells that gave rise to relapse were..."

line 284 language style: "This agree well with our cases which...." probably better if formulated: "This agrees well with our cases, which...."

Lines 303 – 305: Suggest to correct grammar in this sentence: "This demonstrates absence of additional mutational pressure and that an evolutionary early and inherently resistant or dormant cell that have remained unaffected by treatment, caused relapse."

Line 333 – 335: I suggest to rephrase this sentence: "Our results suggest that early KMT2A-r relapse ALL is driven by persisting cells that accumulate mutations, most commonly in TP53 and IKZF1, because of chemotherapy treatment,..." - Leukemia cells don't accumulate mutations because of chemotherapy. Rather, residual cells that have spontaneously acquired TP53 and IKZF1 mutations are selected under the pressure of chemotherapy and then grow out.

Reviewer #3

(Remarks to the Author)

In this study by Ahlgren et al., the authors conducted whole-genome and whole exome sequencing as well as longitudinal analysis on a substantial group of children and infants with relapsing KMT2A-r ALL and AML. Analyses suggest distinct mechanisms of relapse in early relapse of KMT2A-r ALL versus very early ALL relapse and AML relapse at any timepoints. These patterns are differentially characterized by unique somatic mutations linked to drug response and signatures indicative of chemotherapy exposure in early relapse ALL. Despite building upon existing methodologies, the specific focus on KMT2A rearranged leukemias and the scale of the performed analysis confer novelty to the study. Tracking samples over time allows for the identification of clonal dynamics, highlighting the evolution of drug resistance and give a new insightful understanding of the relapsing mechanisms for this form of pediatric leukemias, aiming to provide a groundwork for new therapeutic interventions. Furthermore, the manuscript is well written with a comprehensive set of figures which are easy to follow.

Major points to address:

1. Overall there are several interesting stories being told in this manuscript but it would potentially benefit from some restructuring. Overall, the take home message is that there are different patterns of mutational acquisition in relapse KMT2Ar leukemia (AML v ALL) and in ALL alone (very early v early relapse). The section on longitudinal MRD monitoring via personalized primers is interesting but is somewhat adjacent to the main theme of the paper. Small sample size limits interpretation of the utility of this assay prognostically. However, this approach does yield useful data that supports the initial WGS/WES analyses on relapse samples by showing the acquisition of mutations and the patterns. Thus, perhaps the MRD monitoring could be better in a supplemental figure.

2. Would benefit from a table summarizing what samples were used for what assays/cohorts similar to the supplemental data table but up front in the manuscript to understand how different patients and samples were used.

3. The section on mutational signatures needs to be re-written for more clarity for a reader who may not be familiar with this type of analysis and the prior work. Additional introduction to the approach would be helpful in the results section, not just in the methods.

4. Did any patients suffer lineage switch?

5. Interesting that there were no late relapses in the ALL group. How do the mutations found in the infant cohort in early and very early relapse compare to those found in early and very early relapse ALL without KMT2A?

6. It would be recommended to conduct a functional assay using cell lines (which the author has partially addressed for the PRPS2 mutations) or primary sample to confirm the impact of the frequently identified mutation in drug-response genes TP53 and/or IKZF1 to corroborate the suggested finding that mutation in these genes confers resistance or alters sensitivity to chemotherapy drugs and supporting their role in the observed in the proposed relapse mechanism.

Furthermore, the manuscript could benefit from in vivo experiments using PDX models, especially for the early relapse ALL. This step will enable validation of the finding related to longitudinal monitoring of residual leukemia cell and clonal evolution leading to relapse. By administering various doses of chemotherapy regimens similarly to those used in patient and monitor disease burden it should be possible to measure changes in gene expression and clonal dynamics confirming what identified earlier in patients. Ultimately, if cells from very early relapse are exposed in vitro or in vivo to chemotherapy, would they acquire chemo-related signature?

The last point could potentially be addressed in vitro as well.

Minor:

Figure 1: It will be great to have a schematic for what the author means for early/very early ALL and late relapse in AML before the other panels. This will be helpful for following all the other panels.

Is panel B referring to both AML and ALL? In that case, the author should specify it in the legend.

Same input for figure 2.

Figure 4:

In panel A the author shows the measurable disease at day 15 and 29. What is the rationale behind choosing these two specific timepoints?

Supplementary figure 2 seems to be missing plot H.

Would benefit by a close reading to correct several typos and also could benefit from help from a native English speaker to correct some errors in use, for example "Diagnose" should be "Diagnosis" in Figure 1b.

Reviewer #4

(Remarks to the Author)

Version 1:

Reviewer comments:

Reviewer #1

(Remarks to the Author)

The authors have provided very thorough responses and attention to previous concerns raised, almost all of which appear to be satisfactory.

However, in response to one reviewer request, the authors shared the frequency of study participants in the infant and childhood age categories by ALL vs AML.

The breakdown of 24|2 vs 1|9 shows a highly significant difference in ALL and AML samples by these age categories:

age/type ALL AML
infants 24 2
children 1 9

Fisher's Exact test: $p < 0.000001$

This begs the question of how much of any observed claims comparing ALL and AML relapse pathways (e.g. lines 11-13 of the abstract), could be due to molecular or other changes known to be associated merely with age-related differences within each of these two different acute leukemias separately.

Most of the paper does not dwell on ALL vs AML differences, and so the paper remains a solid contribution in this field.

The authors' point about a lack of certain molecular changes between the diagnosis and relapse samples in AML patients is appreciated. Still, this highly significant potential confounder should at the very least be mentioned in the paper as a potential limitation.

Finally, an additional sentence summarizing (e.g. mean and range) of the ages of the ALL and separately AML patients would make it more clear to the readership

that this bias was known and accounted for there being fewer additional comparisons across ALL and AML. The supplementary table which only lists the ages for children as being ">1 year" would be nice to be replaced by broader age ranges (e.g. 1-5yrs, 5-10yrs, ...).

Overall, this paper is a very nice contribution to the field. Thank you for your work and efforts.

Reviewer #2

(Remarks to the Author)

The authors have addressed all of my points adequately.

Reviewer #3

(Remarks to the Author)

Thank you to the authors for their revised manuscript. Their responses adequately addressed my concerns. While they did not perform all the suggested experiments, they provided thorough explanations, cited relevant literature, and analyzed additional data to support their conclusions. I find their approach reasonable and support the manuscript's progression toward publication.

Reviewer #4

(Remarks to the Author)

REVIEWER COMMENTS

Reviewer #1 (Remarks to the Author):

This is a nice contribution into this rearranged subtype of ALL. Obviously it is a challenge to have a large number of patients, and the authors have done a nice job in presenting their findings, in particular the contrast between mutational signatures in relapse between very early and early patients with ALL, as well as the contrast in changes between diagnosis and relapse in patients with AML. The paper is extremely well written and the investigation is very thorough (especially the additional single cell analysis in some patients to tease out the temporal development of mutations in the longitudinal analyses).

Answer: We thank the reviewer for the very positive comments on our paper.

Only a couple of points that could make the paper completely ready for publication.

1) The patients were accumulated over 18 years, and it should be more clearly stated which samples had bioinformatic analyses performed simultaneously on stored samples and which were performed in near real-time as the sample was obtained. This is important as a multitude of changes can occur over time if samples were not analyzed for mutations simultaneously - from changes in technicians, to changes in software, target sequencing depths, etc. - all of which can affect the ability to detect mutations. The largest finding of 75% vs 0% having TP53 or IKZF1 mutations should be very clear in this respect. If samples were not analyzed under identical circumstances - then efforts to show that such a finding was not due to such changes should be made. E.g., not all samples had a germline sample for a paired tumor-normal analysis. To alleviate concerns that lack of a germline sample in some patients could be influencing the results, an ad-hoc analysis in which only tumor samples from all patients are analyzed identically may show (hopefully) that the contrast between ages is not likely to be artifactual for this TP53/IKZF1 finding.

Answer: None of the patient samples were analyzed in real-time, they were all frozen before analysis as shown in **Supplementary Table 12**, column "Material" [H]. To make it clear with which technologies each sample were analyzed, we have added this information to **Supp. Table 1**, column [O-Q] "WGS", "WES" and "Longitudinal Sequencing". Where D=Diagnosis, G=Germline and R1/2/3/4 = Relapse. Further, we have added the date when the WGS and WES were performed to **Supplementary Table 2** in column [L] "WGS Date" and [U] "WES Date". Data from 6 WES samples was received directly from Agraz-Doblas, A. et al. *Hematologica* 2019, also now indicated in **Supplementary Table 2**. Further, we have added the sampling year of each sample in column [D] "Sampling Year" to make it transparent how long the samples had been stored before analysis, and the Bioinformatics pipeline used for analysis in column [M] and [V]. As already stated in the Materials and Methods, most samples were analyzed with the pipeline developed for the Pediatric Cancer Genome Project (PCGP) at St Judes but some were analyzed with alternative pipelines and we revised the text on pages 20-21, lines 419-437.

Further, in **Supplementary Table 4** we have added column X-Z to show which mutations were called in WES "CalledinWES" [X], WGS CalledinWGS [Y] and or Targeted Sequencing "ValidatedbyTargetedSeq." [Z]. We have also added column a column showing which variants that have been manually inspected in the BAM-files "Manual Review" [S]. For the SVs, in **Supplementary Table 6**, we have added the validation techniques and various support for the event in columns [O-U]. Further, all CNAs were manually inspected and selected genes (*PAX5*, *IKZF1*, *CDKN2A/B*,) were manually inspected by zoomed-in ai plots to ensure that focal events were not missed, as described in the materials and methods.

Of the genes in the ALL and AML heat maps (**Figures 1d and 2b**) 78% (55/72) of the SNVs and 88% (7/8) SVs were validated by another method and remaining alterations were manually inspected. To make this clear and transparent, we have added this information in **Supplementary Table 7** in column [D] "Data" and [O] "IdentifiedBy". We have also separated the GeneName, Amino acid change and Alteration type into three columns for more clarity [A] "Gene", [B] "Achange" and [C] "Alteration".

More specifically, for the genes in the ALL-heat map 79% (31/39) of the SNVs and 86% (6/7) SVs were validated by another method with remaining 8 mutations being manually inspected. Thus, 89% (17/19) SNVs and SVs connected to chemoresistance (*IKZF1*, *TP53*, *CREBBP*, *NT5C2*, *PRPS2*, *WHSC1*) were validated by another method. The two manually inspected SNVs in *TP53* and *WHSC1* were gained at relapse (P121R, *TP53*^{V173L} and P137 *WHSC1*^{E3295K}) and identified by WGS. P121 lacked a GL sample and the *TP53* mutation not identified at diagnosis and present in COSMIC. In addition, we also, as suggested by you in question 5, added the median coverage for the WGS and WES across *TP53* and *IKZF1* in a new table, **Supplementary Table 8** as well as the coverage across these two genes as supplementary Figures (**new Supplementary Fig. 2 and 3**), showing that the genes are sufficiently covered.

To verify that lack of a matched germline sample did not influence our results, we performed a set of additional analyses. In total, 9 out of 33 cases (4/11 v. early ALL cases, and 5/14 early ALL cases) lacked a matched GL sample and were compared to a random high quality GL sample for analysis as described in materials and methods (**Supplementary Table 7, column [I]** now states which samples were analyzed for each case). Of the cases with alterations connected to chemoresistance, 40% (4/10) were lacked a matched GL and all 4 harbored known cancer-associated alterations in *IKZF1* (n=1) and *TP53* (n=4). To make sure the % of alterations were not affected by the lack of a GL sample, we calculated the % of alterations in cases with or without a matched GL sample showing a similar frequency in the two groups (78% (7/9) in cases with a matched GL and 80% (4/5) in cases lacking a matched GL). Of note, 4/5 cases lacking a matched GL had both D-R samples thus, the gain of *IKZF1* and *TP53* alterations at Relapse in three of those cases and their absence at Diagnosis, also support that they are not rare GL events. As stated in the materials and methods section, in cases lacking a matched GL, we only considered alterations if they were present in a recurrent pathway and if they were known cancer-associated and present in COSMIC.

In addition, to further validate our findings, in the revised version, we added an analysis of a cohort of 80 *KMT2A-r* infant patients, from three published studies: Andersson, A.K. et al. *The landscape of somatic mutations in infant MLL-rearranged acute lymphoblastic leukemias* (*Nature Genetics*, 2015), Li, B. et al. *Therapy-induced mutations drive the genomic landscape of relapsed acute lymphoblastic leukemia* (*Blood*, 2020) and Agraz-Doblas, A. et al. *Unraveling the cellular origin and clinical prognostic markers of infant B-cell acute lymphoblastic leukemia using genome-wide analysis* (*Hematologica* 2019). In this analysis, data from 80 infant cases including 18 trios (D-G-R) with 56/80 cases having WGS at diagnosis, 29 also had complementary WES, and 24/80 cases had Affymetrix 6.0 SNP arrays allowing genome-wide CNA, and targeted gene re-sequencing data of 232 genes (including *TP53*, *CREBBP* and *WHSC1* but not *NR3C1*, *NR3C2*, *NT5C2*, *PRPS2*, *PRPS1*, *FPGS*, *MSH2*, *MSH6* and *PMS2*).

Of the 18 trios (6 very early relapse and 12 early relapse), 7/18 cases had D-G-R WGS data, 6/18 WGS at D, but only WES at R, and 5/18 cases had Affymetrix 6.0 plus the 232-genes sequenced. Thus, the patients with complete data allowing all types of alterations to be found were limited but overall show concordance with our results with none of the very early relapse ALL (0/6) having chemo-related changes while 50% of the early relapse ALL had such alterations. However, given that some cases lacked complete genomics data, the % might be underestimated. These results have been added to **page 5, lines 94-99** and as **Supplementary Table 10**. "To validate our findings, we analyzed data from three studies including data from 80 diagnostic *KMT2A-r* infant ALL cases of which 18 had a paired

relapse. This showed that 6/12 (50%) of infants with early relapse, and none of the 6 infants with very early relapse, had such alterations at relapse (Supplementary Table 10). Further, TP53-alterations were rare at diagnosis (2/80 cases), with one or the cases having a very early relapse, but data from the paired relapse was not available."

2) A rather large number of statistical comparisons are performed on this rather small number of patients. Hence, some subset of these findings are likely to be the result of multiple comparisons - and this should be stated as a limitation and that many of these findings likely need labeled as exploratory observations in need of further confirmation. The main hypothesis of the paper seems to have been the contrast in mutational signatures at relapse between very-early and early ALL patients having this rearrangement. Beyond that (or possibly also allowing for the contrast with AML), the other observations are definitely intriguing and hypothesis generating, but should likely all be grouped under this explorative caveat.

Answer: We acknowledge that the number of patients is small which could influence the results and statistics, and it has been a struggle to get hold of these rare samples. In the revised version of the paper, we have included an extended analysis of published data of 80 *KMT2A-r* infants including 18 cases with a paired relapse, supporting our findings (see page 5 in the revised version of the manuscript and new **Supplementary Table 10**). As described in our reply to you in question #1, similar to our data, there was a paucity of acquired alterations in chemoresistance-associated genes in very ALL relapse while they are common at early relapse (50%), although the frequency was somewhat lower than ours at 85%. With that said, also the validation cohort is also quite small with only 6 very early and 12 early relapse cases, and not all cases had complete genomics data which could lower the true % (our own data includes 11 very early and 14 early cases), also making the % in the validation cohort prone to skewing in one or the other direction. Importantly, to make it clear to the reader that our study has these limitations, we have added a statement in the discussion on **pages 14 lines 274-278**: *"To validate our findings, we analyzed data from 18 KMT2A-r ALL trios, showing a frequency of chemo-associated lesion on a par with our data in early relapse cases (50%, 6/12), and a paucity of such changes in very early relapse cases (0/6). Still, the number of patients analyzed remain low, and some lacked complete genomics data, thus larger studies are needed to confirm our results."*

Further, we also added a statement in the MRD-discussion section on **page 16, lines 339-340** *"However, the limited cohort size may affect the results, highlighting the need for validation in larger studies."*

minor:

1) in the first paragraph, please provide a directly comparable contrast between general and infant *KMT2A-r* ALL (either both EFS or both OS).

Answer: Thank you for this comment. We have modified our sentence about the survival accordingly (bold below) in the main text, **page 2, line 19**: *However, ALL in infants, i.e., children aged 0-12 months, with KMT2A-rearrangements (KMT2A-r), still have a dismal prognosis with a 21-45% 6-year event-free survival (EFS) rate compared to 74% EFS in non-KMT2A-r ALL.*

2) Lines 51-53, please make it easier on the reader to tell how many with ALL and AML were infants and children (still can't tell from figure 1 easily).

Answer: Thank you for pointing this out. We have added this accordingly in the main text **page 4, lines 52-53 (bold)** *"To gain insights into mechanisms of relapse, we performed WGS and WES on 36 cases of relapsed KMT2A-r ALL (n=25) or AML (n=11) including 26 infants (24 ALL, 2 AML) and 10 children (1 ALL, 9 AML) (average coverage: WGS 43X, WES 140X, Fig. 1a,b, Supplementary Fig. 1a,b, and*

Supplementary Tables 1,2)” and updated Figure 1a with the number of infants and children, and the number of cases with v. early, early, and late relapse.

3) Line 555 the single word Invitrogen needs removed

Answer: “Invitrogen” was removed, thank you for picking this up.

4) Somewhere in the introduction it may help the readership to clearly state the cohort selection criteria being based on KMT2A rearrangement at diagnosis as opposed to at relapse, and to indicate the percent of patients having detectable KMT2A rearrangements at relapse.

Answer: As stated on lines 34-36 in the introduction, all patients were *KMT2A-r* at both diagnosis and relapse and all had detectable *KMT2A-r* at both timepoints. For clarity, we have renamed column [D] from FusionGene to FusionGeneatDiagnosis and added column FusionGeneatRelapse [E] in **Supplementary Table 1**.

5) The bioinformatic section of the Methods there should be at least some summary statistics of the depths of sequencing obtained with the different techniques. Especially nice if that information was included specifically for depth of coverage of the drug-response genes - perhaps even with a supplemental figure showing that for all samples and regions of those genes (at least for TP53 and IKZF1).

Answer: We have added the average sequencing depth of the WGS/WES data to the **1st paragraph on page 4** in the manuscript: “*To gain insights into mechanisms of relapse, we performed WGS and WES on 36 cases of relapsed KMT2A-r ALL (n=25) or AML (n=11) including 26 infants (24 ALL, 2 AML) and 10 children (1 ALL, 9 AML) (average coverage: WGS 43X, WES 141X, Fig. 1a,b, Supplementary Fig. 1a,b, and Supplementary Tables 1,2)*”. The average coverage per sample and its matched germline for WGS and WES is indicated in **Supplementary Table 2** and **Supplementary Figures 1a,b**, and for the longitudinal sequencing, the coverage can be found in **Supplementary Table 12**. Further, as suggested, we have added new **Supplementary Figures 2-3** showing the WGS/WES coverage for *TP53* and *IKZF1* as well as a new **Supplementary Table 8** showing the median coverage across *TP53* and *IKZF1*.

Reviewer #2 (Remarks to the Author):

In the current manuscript, the authors present a comprehensive exploratory NGS based genetic analysis of infant and childhood KMT2A-rearranged acute leukemia. By sequencing serial time-point samples of 33 ALL and AML patients with WGS, WES and concomitant deep sequencing they not only provide an important detailed description of the complex genetic and clonal landscape of this rare and aggressive subentity of ALL/AML, but also make some interesting novel discoveries. These are namely that very early versus early forms display differential mutational patterns, which suggest different biologic mechanisms of relapse, i.e. early ALL relapse is characterized by enriched acquisition of novel and frequently combined TP53 and IKZF1 mutations, while the analyzed cohort of very early ALL relapse is void of these mutations. Moreover, while relapsed ALL cases have an enrichment of chemotherapy-induced signatures, AML did not contain these signatures, also suggesting inherent differences of relapse between these two entities. The manuscript is well written and structured. The data has been thoroughly analyzed and interpreted and provides an important database for further research in these leukemia entities.

Answer: We thank the reviewer for the very positive comments on our paper.

There are some **major** suggestions for improvement of the manuscript:

1. The strongest and most important discovery in this work are the differential mutational profiles between very early ALL relapse versus the early relapse cohort, which supports the hypothesis formulated by authors that early relapse is possibly driven by selection of cells with TP53 + IKFZ1 mutations under the pressure of chemotherapy as opposed to an underlying primary / inherent resistance in very early relapse. However, the underlying data set is so far only descriptive and despite respectful acknowledgement of the collection of these rare primary samples, which has taken many years, the sample set (n=9 very early and 12 early, Fig. 1d) is still quite low to rank this as a strong conclusion but rather limits it to a strongly suggestive hypothesis. The manuscript would gain great improvement if the investigators were able to support this observation by some sort of further confirmation, which could e.g. either be by a targeted validation (i.e. not necessarily WES or WGS) of this observation in another cohort, or by functional validation of the hypothesis in corresponding pre-clinical models e.g. by showing that KMT2A-rearranged samples with (e.g. engineered) TP53 + IKFZ1 mutations are more resistant to the applied therapies than original samples without these mutations. – Or if these preferable approaches are not feasible, at least an evaluation by a statistician of the data in Figure 1d and a critical discussion, that the evidence is limited by missing functional data and low sample numbers.

Answer: First, with regards to the possibility of validating our findings, it has overall, been a struggle to get hold of these rare samples, in particular of paired relapse samples. During the revision we tried to get additional cases through St Jude and Princess Maxima which resulted in new data on 2 resistant and 2 early relapse cases from St Judes. We thus increased the number of very early/resistant cases from 9 to 11 cases, and early relapse from 12 to 14 cases. All data and figures have been updated accordingly. The data remain similar, but now with 9% of very early relapse ALL having chemo-associated changes and 79% of early relapse (before 0 and 83%) and the ms have been updated accordingly.

Since getting additional cases were so difficult, to further validate our findings, we collected published cases from the literature, and have added an analysis of external data from a cohort of 80 KMT2A-r infant patients, from three studies: Andersson, A.K. et al. *The landscape of somatic mutations in infant MLL-rearranged acute lymphoblastic leukemias* (*Nature Genetics*, 2015), Li, B. et al. *Therapy-induced mutations drive the genomic landscape of relapsed acute lymphoblastic leukemia* (*Blood*, 2020) and Agraz-Doblas, A. et al. *Unraveling the cellular origin and clinical prognostic markers of infant B-cell acute lymphoblastic leukemia using genome-wide analysis* (*Hematologica* 2019). Specifically, we analyzed data from 80 infant cases including 18 trios (D-G-R) with 56/80 cases having WGS at diagnosis and 29 also had complementary WES, and 24/80 cases had Affymetrix 6.0 SNP arrays allowing for genome-wide CNA and targeted gene re-sequencing data of 232 genes (including *TP53*, *CREBBP* and *WHSC1* but not *NR3C1*, *NR3C2*, *NT5C2*, *PRPS2*, *PRPS1*, *FPGS*, *MSH2*, *MSH6* and *PMS2*).

Of the 18 trios (6 very early and 12 early), 7/18 cases had complete WGS data, 6/18 WGS at D and only WES at R, and 5/18 cases had Affymetrix 6.0 plus the 232-genes sequenced at D-R. Thus, the patients with complete data allowing all types of alterations to be found were limited but overall show concordance with our results with none of the very early relapse cases (0/6) having chemo-related changes while 50% of the early relapse cases had such alterations. Given that some cases lacked complete genomics data, the % might be underestimated. These results have been added in a new **Supplementary Table 10**, as well as to **page 5, lines 94-99**. “To validate our findings, we analyzed data from three studies, including data from 80 diagnostic KMT2A-r infant ALL cases of which 18 had a paired relapse. This showed that 6/12 (50%) of infants with early relapse, and none of the 6 infants with very early relapse, had such alterations (**Supplementary Table 10**). Further, TP53-alterations were rare at diagnosis (2/80 cases), with one of the cases having a very early relapse, but data from the paired relapse was not available. “

We also revised the discussion to include the limitations of our data as well as of the validation cohort, see **page 13-14, lines 274-278**. *“To validate our findings, we analyzed data from 18 KMT2A-r ALL trios, showing a frequency of chemo-associated lesion on a par with our data in early relapse cases (50%, 6/12), and a paucity of such changes in very early relapse cases (0/6). Still, the number of patients analyzed remain low, and some lacked complete genomics data, thus larger studies are needed to confirm our results.”*

With regards to functional validation (also suggested by reviewer 3, #6), several papers have addressed the impact of *TP53* and *IKZF1* alterations on chemoresistance. With regards to *TP53*, Li, B. et al. *Therapy-induced mutations drive the genomic landscape of relapsed acute lymphoblastic leukemia (Blood, 2020)* who did functional validation of two *TP53* mutations (R248Q and R196G), showing resistance to idarubicin and vincristine. Demir et al, *Therapeutic targeting of mutated p53 in pediatric acute lymphoblastic leukemia (Haematologica, 2020)* showed that *TP53* mutated cases are insensitive to doxorubicin and that the small molecule inhibitor APR-246 restored *TP53* function and APR-246 was also found to synergize with doxorubicin in a PDX model with mutant *TP53*. In Oshima K., et al., *Mutational and functional genetics mapping of chemotherapy resistance mechanisms in relapsed acute lymphoblastic leukemia (Nature Cancer, 2020)* they used CRISPR-Cas9 gene knockout of *TP53* in leukemic cell lines showing that *TP53* loss resulted in increased resistance to doxorubicin, cytarabine and vincristine. In Yang et al, *Chemotherapy and mismatch repair deficiency cooperate to fuel TP53 mutagenesis and ALL relapse (Nature Cancer, 2021)*, they demonstrate that *TP53*^{R248Q} in relapsed ALL originates through synergistic mutagenesis from thiopurine treatment and MMR deficiency. Finally, Cox WPJ., et al, *Histone deacetylase inhibition sensitizes p53-deficient B-cell precursor acute lymphoblastic leukemia to chemotherapy (Haematologica, 2024)*, recently showed that CRISPR/Cas9-induced loss of *TP53* drives resistance to a large majority of drugs used to treat relapsed ALL.

With regards to *IKZF1*, Scheijen, B. et al. *Tumor suppressors BTG1 and IKZF1 cooperate during mouse leukemia development and increase relapse risk in B-cell precursor acute lymphoblastic leukemia patients (Haematologica, 2017)* showed that *IKZF1* deletions lead to increased glucocorticoid resistance in haplodeficient *IKZF1* mice, and the same has been shown in primary patients (Marke R, et al, *Leukemia 2016*). Rogers, J. H. et al *Modeling IKZF1 lesions in B-ALL reveals distinct chemosensitivity patterns and potential therapeutic vulnerabilities (Blood, 2021)* assessed *IKZF1* deletions using CRISPR-Cas9 targeting exon 2 or 3 and demonstrated increased resistance to dexamethasone, asparaginase, and daunorubicin as compared to the control. Further, Vervoort, B et al. *IKZF1 gene deletions drive resistance to cytarabine in B-cell precursor acute lymphoblastic leukemia, showed that loss of IKZF1 in cell lines and PDX-models is associated with Cytarabine resistance (Haematologica, 2024)*.

Combined, numerous studies support that *TP53* and *IKZF1* alterations lead to chemoresistance in agreement with our own genetic data on primary patient material. We thank the reviewer for bringing this up and we have revised our discussion on **page 14 lines 278-281** and added the references above *“In agreement with the patient data, functional studies have demonstrated that alterations in TP53 and IKZF1 drives resistance to numerous cytostatics, suggesting that alternative treatments are needed to successfully treat these patients.”*

Further, we do agree that studying this using PDX models combined with chemotherapy-treatment indeed would be very elegant, but we feel that it out of the scope of the present study and would be best fitted as a follow-up study.

Minor:

line 44: language style: „...cells that gave rise to relapse was...”: probably better if formulated: “...cells that gave rise to relapse were...”

Answer: Thanks for finding this, we have changed it accordingly.

line 284 language style: “This agree well with our cases which....” probably better if formulated: “This agrees well with our cases, which....”

Answer: A comma has been added as suggested.

Lines 303 – 305: Suggest to correct grammar in this sentence: “This demonstrates absence of additional mutational pressure and that an evolutionary early and inherently resistant or dormant cell that have remained unaffected by treatment, caused relapse.”

Answer: This sentence has been rewritten to: “*This demonstrates **the** absence of additional mutational pressure and that an evolutionarily early and inherently resistant or dormant cell, which remained unaffected by treatment, caused **the** relapse.*” **Lines 314-316**

Line 333 – 335: I suggest to rephrase this sentence: “Our results suggest that early KMT2A-r relapse ALL is driven by persisting cells that accumulate mutations, most commonly in TP53 and IKZF1, because of chemotherapy treatment,...” - Leukemia cells don’t accumulate mutations because of chemotherapy. Rather, residual cells that have spontaneously acquired TP53 and IKZF1 mutations are selected under the pressure of chemotherapy and then grow out

Answer: Our data and several previously published studies (Li, B. et al. *Therapy-induced mutations drive the genomic landscape of relapsed acute lymphoblastic leukemia* (Blood, 2020), Szkriszt, B. et al. *A comprehensive survey of the mutagenic impact of common cancer cytotoxics* (Genome Biology 2016), Yang F. et al., *Chemotherapy and mismatch repair deficiency cooperate to fuel TP53 mutagenesis in ALL* (Nature Cancer 2021), and Yu, S-L. et al. *FPGS relapse-specific mutations in relapsed childhood acute lymphoblastic leukemia* (Scientific Reports, 2020), van der Ham et al, *Mutational mechanisms in multiple relapsed pediatric acute lymphoblastic leukemia* (Leukemia, 2024), indicate that resistance can be driven by both preexisting resistant leukemic cells or by persistent cells acquiring resistance-mutations likely because of the treatment itself. In Li, B. et al. they showed this by mathematical modeling and mutational signature analysis, as well as by in-vitro drug treatment which re-capitulated the chemotherapy-induced signature observed in patients and that characterized many of the drug-induced mutations, in Yang F. et al., it was shown that thiopurine treatment in mismatch-repair deficient leukemias induces TP53 R248Q, in Szkriszt, B. et al by cisplatin-treated cell cultures which increases mutagenesis, and lastly in Yu, S-L. et al. by sequencing of diagnostic and relapse samples in 372 childhood ALL cases where the *FPGS*, *PRPS1* and *NT5C2* was not identified by deep-sequencing at diagnosis. In van der Ham et al, they show that thiopurine exposure was the most prominent source of new mutations in relapse, affecting over half of the studied patients in first and/or later relapse and causing potential relapse-driving mutations in multiple patients.

In addition to above studies, the failure of detection of relapse-specific mutations in genes with a role in drug-resistance already at diagnosis has been demonstrated in additional studies, including with higher-resolution sequencing methods such as error-corrected sequencing and dd-PCR ((Tzoneva et al., Nature 2018, Pilheden et al, HemaSphere 2022). In Tzoneva et al, it was shown that *NT5C2* mutations can be detected in complete remission samples prior to relapse, however, if present in the clonal repertoire at diagnosis, they represent minor populations below the sensitivity of molecular assays (Tzoneva et al., Nature 2018). The absence of these mutations at diagnostic

together with recent data that showing that many of the mutations found at relapse are marked by mutational signatures that occur in chemotherapy-treated cells, suggest that these drugs damage cells in such a way that of cells persist treatment, many of the new mutations will be a result of the treatment itself.

Thus, emerging evidence suggest that at least some driver mutations may be induced by treatment, rather than being pre-existing and selected for during treatment. In line with this, we failed to detect the drug-induced mutations at diagnosis with deep-sequencing, as well as with error-corrected sequencing (*TP53*) in one of the cases (Pilheden et al, HemaSphere 2022). In contrast to *TP53* and other drug-associated mutation (*PRPS2*, *NT5C2*), *IKZF1* deletions may be pre-existing and selected for, as in 2/8 cases, the deletions were indeed present at diagnosis with one identified by PCR and thus below the resolution of WGS (P58). Nevertheless, we would like to suggest the possibility that at least some of the drug-induced mutations are acquired because of treatment and then selected for as suggested by the data. We have, however, re-written the final paragraph on pages 16/17 to make it more open and to suggest this possibility, and we have also added above references to support our discussion on page 15. "Page 16.....*Our results suggest that some of the mutations that drive early KMT2A-r relapse ALL accumulate in persisting leukemia cells because of chemotherapy treatment, while very early relapse ALL is driven by an inherent resistance.*"

Reviewer #3 (Remarks to the Author):

In this study by Ahlgren et al., the authors conducted whole-genome and whole exome sequencing as well as longitudinal analysis on a substantial group of children and infants with relapsing KMT2A-r ALL and AML. Analyses suggest distinct mechanisms of relapse in early relapse of KMT2A-r ALL versus very early ALL relapse and AML relapse at any timepoints. These patterns are differentially characterized by unique somatic mutations linked to drug response and signatures indicative of chemotherapy exposure in early relapse ALL. Despite building upon existing methodologies, the specific focus on KMT2A rearranged leukemias and the scale of the performed analysis confer novelty to the study. Tracking samples over time allows for the identification of clonal dynamics, highlighting the evolution of drug resistance and give a new insightful understanding of the relapsing mechanisms for this form of pediatric leukemias, aiming to provide a groundwork for new therapeutic interventions. Furthermore, the manuscript is well written with a comprehensive set of figures which are easy to follow.

Answer: We thank the reviewer for the very positive comments on our paper.

Major points to address:

1. Overall there are several interesting stories being told in this manuscript but it would potentially benefit from some restructuring. Overall, the take home message is that there are different patterns of mutational acquisition in relapse KMT2Ar leukemia (AML v ALL) and in ALL alone (very early v early relapse). The section on longitudinal MRD monitoring via personalized primers is interesting but is somewhat adjacent to the main theme of the paper. Small sample size limits interpretation of the utility of this assay prognostically. However, this approach does yield useful data that supports the initial WGS/WES analyses on relapse samples by showing the acquisition of mutations and the patterns. Thus, perhaps the MRD monitoring could be better in a supplemental figure.

Answer: We understand your concern regarding the small sample size and have discussed whether this section should be moved to the supplementary or not. However, we do believe that the analysis adds extra support to our previous findings that very early ALL relapse is connected to a more aggressive disease, not responding as rapidly and efficiently to treatment, and thus that it should be

kept. We have, however, shortened the section significantly (from 459 words to 341 words) and have also added a statement in the discussion that the results have limitations due to the small sample size. **Page 16, lines 339-340, “However, the limited cohort size may affect the results, highlighting the need for validation in larger studies.”** Further, to make the data easier to interpret, we have added column [J] “InCRd15” and [W] “InCRd29”, stating whether the patients were in clinical remission (CR) at MRD-timepoint 1 and 2.

2. Would benefit from a table summarizing what samples were used for what assays/cohorts similar to the supplemental data table but up front in the manuscript to understand how different patients and samples were used.

Answer: Thank you for this suggestion. We have added this information in the introduction at page 2, **line 38:** *In addition, we performed targeted deep-sequencing of 258 samples from 30 cases including 14 of the relapse cohort, during the disease course.* Further, for each patient, we have added columns O-P-Q stating if WGS, WES or longitudinal seq were performed, in **Supplementary Table 1**, and we have also modified the **Main Figure 1 and 2** and associated legends to make it easier to follow. We hope this is more transparent and clearer now.

3. The section on mutational signatures needs to be re-written for more clarity for a reader who may not be familiar with this type of analysis and the prior work. Additional introduction to the approach would be helpful in the results section, not just in the methods.

Answer: To clarify this we have re-written the introductory text in that section from “To understand the history of the cells seeding relapse, we examined mutational signatures based on trinucleotide context...” to **“Based on the trinucleotide context, 60 single-base substitution (SBS) signatures have been identified and attributed to both known and unknown etiologies¹⁶ To understand which mutational processes that were active, we examined the mutational signatures at relapse (lines 145-147, page 7).”**

We have also added some text and modified the text in the materials and methods section **lines 449-453.** *“The R-package MutationalPatterns was utilized to assess the relative contribution of all single base substitutions in non-repetitive regions identified through whole-genome sequencing (WGS) to the Cosmic SBS mutational profiles. Each signature was classified as Clock-like, APOBEC/AID, Mismatch Repair (MMR), UV, Chemotherapy treatment, Biological, ROS, Tobacco, Chemical exposure or Unknown based on their description in Cosmic, and each patient was assigned based on their most common classification to their “primary signature” (Supplementary Table 11). When calculating the average VAF per signature-classification, each mutation was first assigned to the most likely signature: All 96 possible motifs were designated a most likely signature, by mapping each one to the Cosmic SBS mutational signature where the motif has the highest contribution.”*

4. Did any patients suffer lineage switch?

Answer: To our knowledge none of the patients had a lineage switch, they had the same disease at diagnosis and relapse

5. Interesting that there were no late relapses in the ALL group. How do the mutations found in the infant cohort in early and very early relapse compare to those found in early and very early relapse ALL without KMT2A?

Answer: According to the Interfant-06 trial, including 651 infants (both KMT2A and non-KMT2A), 90% of relapses occurred within two years of treatment and 66% relapsed during the first year after

diagnosis. This underlines the aggressivity in infant leukemia. To make this more clear, we revised a sentence in the main, **rows 27-28** to read “Over 90% of infants reach clinical remission (CR), but are prone to rapid relapse, with 90% of relapses occurring within 2 years from diagnosis and 66% within one year, and our understanding of the mechanisms driving relapse remains limited.”

Further, a large study by Li, B. et al. *Therapy-induced mutations drive the genomic landscape of relapsed acute lymphoblastic leukemia (Blood, 2020)*, included data on 103 pediatric D-G-R ALL cases, including 8 infants (5 *KMT2A-r*) and 95 children (4 *KMT2A-r*) whereof 24 had a very early relapse, 57 an early relapse and 22 cases a late relapse. Six of the *KMT2A-r* cases had a very early relapse and three an early relapse, thus no late relapses in the St Jude cohort either.

To determine how the frequency of mutations in our infant *KMT2A-r* cohort compared to those found in early and very early relapse non-*KMT2A-r* ALL cases, we analyzed the data from above paper which included 18 non-*KMT2A-r* very early relapse cases (1 infant and 17 children) and 54 non-*KMT2A-r* early relapse cases (1 infant and 53 children). This analysis showed that 22% (4/18) of very early relapse ALL had relapse-specific mutations in genes connected to chemoresistance: *IKZF1* (n=1), *CREBBP* (n=1), *NT5C2* (n=1), and *FPGS* (n=1) as compared 0% (0/9 infants) in our study. Of note, one of their *KMT2A-r* childhood cases with very early relapse had a mutation in *WHSC1* (17%, 1/6) suggesting a possible mutational difference between *KMT2A-r* children and infants since no *WHSC1* mutations were identified in our infant cases with very early relapse. In early relapse non-*KMT2A-r* ALL (1 infant and 53 children), 70% (38/54) had relapse-specific mutations in genes connected to chemoresistance (*NR3C1*, *NR3C2*, *IKZF1*, *TP53*, *CREBBP*, *NT5C2*, *PRPS2*, *PRPS1*, *FPGS*, *MSH2*, *MSH6*, *PMS2* and *WHSC1*), as compared to 83% (10/12, 11 infants and 1 child) in our study. This comparison has been added to the discussion on page 13 at **line 267** “.....and restricting the analyses to non-*KMT2A-r* cases, in 22% and 70%, respectively.

Importantly, none of the 94 non-*KMT2A-r* relapse cases (very early, early and late relapses) in the study by Li B et al., had both an *IKZF1* and a *TP53* alteration at relapse, suggesting that this is unique to *KMT2A-r* infant ALL. To make this clearer, we have revised the text in the discussion on **page 13, lines 268-269** “**Importantly, co-existing *IKZF1* and *TP53* alterations at relapse were not seen in any of these cases, suggesting that this may be unique to *KMT2A-r* ALL.**”

6. It would be recommended to conduct a functional assay using cell lines (which the author has partially addressed for the *PRPS2* mutations) or primary sample to confirm the impact of the frequently identified mutation in drug-response genes *TP53* and/or *IKZF1* to corroborate the suggested finding that mutation in these genes confers resistance or alters sensitivity to chemotherapy drugs and supporting their role in the observed in the proposed relapse mechanism.

Answer: Several papers have addressed the impact of *TP53* and *IKZF1* alterations on chemoresistance. With regards to *TP53*, Li, B. et al. *Therapy-induced mutations drive the genomic landscape of relapsed acute lymphoblastic leukemia (Blood, 2020)* who did functional validation of two *TP53* mutations (R248Q and R196G), showing resistance to idarubicin and vincristine. Demir et al, *Therapeutic targeting of mutated p53 in pediatric acute lymphoblastic leukemia (Haematologica, 2020)* showed that *TP53* mutated cases are insensitive to doxorubicin and that the small molecule inhibitor APR-246 restored *TP53* function and APR-246 was also found to synergize with doxorubicin in a PDX model with mutant *TP53*. In Oshima K., et al., *Mutational and functional genetics mapping of chemotherapy resistance mechanisms in relapsed acute lymphoblastic leukemia (Nature Cancer, 2020)* they used CRISPR-Cas9 gene knockout of *TP53* in leukemic cell lines showing that *TP53* loss resulted in increased resistance to doxorubicin, cytarabine and vincristine. In Yang et al, *Chemotherapy and mismatch repair deficiency cooperate to fuel TP53 mutagenesis and ALL relapse (Nature Cancer, 2021)*, they demonstrate that *TP53*^{R248Q} in relapsed ALL originates through synergistic mutagenesis from

thiopurine treatment and MMR deficiency Finally, Cox WPJ., et al, *Histone deacetylase inhibition sensitizes p53-deficient B-cell precursor acute lymphoblastic leukemia to chemotherapy* (Haematologica, 2024), recently showed that CRISPR/Cas9-induced loss of TP53 drives resistance to a large majority of drugs used to treat relapsed ALL.

With regards to *IKZF1*, Scheijen, B. et al. *Tumor suppressors BTG1 and IKZF1 cooperate during mouse leukemia development and increase relapse risk in B-cell precursor acute lymphoblastic leukemia patients* (Haematologica, 2017) showed that *IKZF1* deletions lead to increased glucocorticoid resistance in haplodeficient *IKZF1* mice, and the same has been shown in primary patients (Marke R, et al, *Leukemia* 2016). Rogers, J. H. et al *Modeling IKZF1 lesions in B-ALL reveals distinct chemosensitivity patterns and potential therapeutic vulnerabilities* (Blood, 2021) assessed *IKZF1* deletions using CRISPR-Cas9 targeting exon 2 or 3 and demonstrated increased resistance to dexamethasone, asparaginase, and daunorubicin as compared to the control. Further, Vervoort, B et al. *IKZF1 gene deletions drive resistance to cytarabine in B-cell precursor acute lymphoblastic leukemia*, showed that loss of *IKZF1* in cell lines and PDX-models is associated with Cytarabine resistance (Haematologica, 2024).

Combined, numerous studies support that *TP53* and *IKZF1* alterations lead to chemoresistance in agreement with our own genetic data on primary patient material. We thank the reviewer for bringing this up and we have revised our discussion on **page 14, lines 278-281** and added the references above **“In agreement with the patient data, functional studies have demonstrated that alterations in *TP53* and *IKZF1* drives resistance to numerous cytostatics, suggesting that alternative treatments are needed to successfully treat these patients.”**

Furthermore, the manuscript could benefit from in vivo experiments using PDX models, especially for the early relapse ALL. This step will enable validation of the finding related to longitudinal monitoring of residual leukemia cell and clonal evolution leading to relapse. By administrating various doses of chemotherapy regimens similarly to those used in patient and monitor disease burden it should be possible to measure changes in gene expression and clonal dynamics confirming what identified earlier in patients. Ultimately, if cells from very early relapse are exposed in vitro or in vivo to chemotherapy, would they acquire chemo-related signature?

The last point could potentially be addressed in vitro as well.

Answer: We do agree that studying this using PDX models would indeed be very elegant, but we feel that it out of the scope of the present study and would be best fitted as a follow-up study. With regards to the last comment, there are some evidences already for this, for example, in Levantić J. et al *Mutational signatures are markers of drug sensitivity of cancer cells* (Nature Communications, 2022), cell lines were analyzed, and the authors show that different drugs gave rise to certain signatures. Further, in Li, B. et al. *Therapy-induced mutations drive the genomic landscape of relapsed acute lymphoblastic leukemia* (Blood, 2020) two novel mutational signatures connected to chemoresistance were identified and crossed checked across 1889 diagnostic cancer samples, spanning 36 cancer types, showing that the signatures were found in ALL relapse samples and pointed towards Thiopurine treatment which is primarily given to ALL, but not to those with other cancers. This was then experimentally validated in MCF10A cells (non-cancerous) that received treatment with Thiopurines which resulted in a mutational spectrum that resembled the sought for signature. Further, 46% of the acquired resistance mutations in *NT5C2*, *PRPS1*, *NR3C1*, and *TP53* were marked by the identified signatures, suggesting that they were indeed a result of treatment. We hope that the revisions made in the discussion can make up for the lack of new PDX data.

Minor:

Figure 1: It will be great to have a schematic for what the author means for early/very early ALL and late relapse in AML before the other panels. This will be helpful for following all the other panels.

Answer: Thank you for this comment. We have added such a description in the figure legend, and we have also stated the definition of relapse time in all figures in panel 1 and 2 in Figure 1, e.g. Very early (<9m after diagnosis CR) and Early (>9m after diagnosis) for all and Early (<1y in CR) and Late (>1y in CR). We hope that this makes it easier to follow.

Is panel B referring to both AML and ALL? In that case, the author should specify it in the legend.

Answer: Yes, in Figure 1B, the cohorts are combined. We have added the number of ALL and AML samples investigated with the different technologies in the figure and the figure legend to make this clear.

Same input for figure 2.

Answer: We have added the definition of relapse time in the figure legend as well as to Figure 2 a-c.

Figure 4: In panel A the author shows the measurable disease at day 15 and 29. What is the rationale behind choosing these two specific timepoints?

Answer: Measurable residual disease is measured clinically in ALL at days 15,29, 72 and 90, and since day 29 have been shown to have clinical impact with regards to pin-point cases with a high risk of relapse, we focused on the early time points. This has been clarified on **page 10, line 205**: "*At day 15, the first MRD-measurement during induction therapy,*".

Supplementary figure 2 seems to be missing plot H.

Answer: Thanks for finding this, it has been corrected.

Would benefit by a close reading to correct several typos and also could benefit from help from a native English speaker to correct some errors in use, for example "Diagnose" should be "Diagnosis" in Figure 1b.

Answer: Thanks for pointing this out, it has been corrected in the manuscript, all figures and supplementary tables. Further, the manuscript has also been submitted to Grammarly to find such errors.

Reviewer #4 (Remarks to the Author):
